# Heterogeneous Multi-player Multi-armed Bandits: Closing the Gap and Generalization

**Chengshuai Shi**
University of Virginia
cs7ync@virginia.edu

**Wei Xiong**
The Hong Kong University of Science and Technology
wxiongae@connect.ust.hk

**Cong Shen**
University of Virginia
cong@virginia.edu

**Jing Yang**
The Pennsylvania State University
yangjing@psu.edu

## Abstract

Despite the significant interests and many progresses in decentralized multi-player multi-armed bandits (MP-MAB) problems in recent years, the regret gap to the natural centralized lower bound in the heterogeneous MP-MAB setting remains open. In this paper, we propose BEACON – *Batched Exploration with Adaptive COmmunicatioN* – that closes this gap. BEACON accomplishes this goal with novel contributions in implicit communication and efficient exploration. For the former, we propose a novel adaptive differential communication (ADC) design that significantly improves the implicit communication efficiency. For the latter, a carefully crafted batched exploration scheme is developed to enable incorporation of the combinatorial upper confidence bound (CUCB) principle. We then generalize the existing *linear-reward* MP-MAB problems, where the system reward is always the sum of individually collected rewards, to a new MP-MAB problem where the system reward is a general (nonlinear) function of individual rewards. We extend BEACON to solve this problem and prove a logarithmic regret. BEACON bridges the algorithm design and regret analysis of combinatorial MAB (CMAB) and MP-MAB, two largely disjointed areas in MAB, and the results in this paper suggest that this previously ignored connection is worth further investigation.

## 1   Introduction

Motivated by the application of cognitive radio (Anandkumar et al., 2010, 2011; Gai et al., 2010), the multi-player version of the multi-armed bandits problem (MP-MAB) has sparked significant interests in recent years. MP-MAB takes player interactions into account by having multiple decentralized players simultaneously play the bandit game and interact with each other through arm collisions.

Prior MP-MAB studies mostly focus on the homogeneous variant, where the bandit model is assumed to be the same across players (Liu and Zhao, 2010; Rosenski et al., 2016; Besson and Kaufmann, 2018). Recent attentions have shifted towards the more general MP-MAB model with player-dependent bandit instances (i.e., the *heterogeneous* variant) (Bistritz and Leshem, 2018, 2020; Tibrewal et al., 2019; Boursier et al., 2020). However, unlike the homogeneous variant, the current understanding of the heterogeneous setting is still limited.

- Recent advances (Boursier and Perchet, 2019; Wang et al., 2020) show that for the homogeneous setting, decentralized MP-MAB algorithms can achieve almost the same performance as centralized ones. However, state-of-the-art results in heterogeneous variants still have significant gaps from

35th Conference on Neural Information Processing Systems (NeurIPS 2021).

the centralized performance. It remains an open problem whether a decentralized algorithm can approach the centralized performance for the heterogeneous MP-MAB variant.
- All prior MP-MAB works are confined to a *linear* system reward function: the system reward is the sum of individual outcomes from players. However, practical system objectives are often captured by more complicated nonlinear reward functions, e.g., the minimal function (see Section 5.1).

In this paper, we make progress in the aforementioned problems for decentralized heterogeneous MP-MAB. A novel algorithm called BEACON – Batched Exploration with Adaptive COmmunicatioN – is proposed and analyzed. In particular, this work makes the following contributions.

- BEACON introduces several novel ideas to the design of implicit communication and efficient exploration. For the former, a novel adaptive differential implicit communication (ADC) scheme is proposed, which can significantly lower the implicit communication loss compared to the state of the art. For the latter, core principles from CUCB (Chen et al., 2013) are incorporated with a batched exploration design, which leads to both efficient and effective explorations.
- For the linear reward function, we rigorously show that regret bounds of BEACON, both problem-dependent and problem-independent, not only improve all prior regret analyses but more importantly are capable of *approaching the centralized lower bounds*, thus answering the aforementioned open problem positively.
- We then propose to generalize the study of heterogeneous MP-MAB to general (nonlinear) reward functions. BEACON is extended to solve such problems and we show that it achieves a regret of $O(\log(T))$, where $T$ is the time horizon. The analysis itself holds important value as it bridges the regret analysis of combinatorial MAB (CMAB) and MP-MAB.
- BEACON achieves impressive empirical results. It not only outperforms existing decentralized algorithms significantly, but indeed has a comparable performance as the centralized benchmark, hence corroborating the theoretical analysis. Remarkably, BEACON with the linear reward function generally achieves $\sim 6\times$ improvement over the state-of-the-art METC (Boursier et al., 2020).

## 2 Problem Formulation

A decentralized MP-MAB model consists of $K$ arms and $M$ players. As commonly assumed (Bistritz and Leshem, 2020; Boursier et al., 2020), there are more arms than players, i.e., $M \leq K$, and initially the players have knowledge of $K$ but not $M$. Furthermore, no *explicit* communications are allowed among players, which results in a decentralized system. Also, time is assumed to be slotted, and at time step $t$, each player $m \in [M]$ chooses and pulls an arm $s_m(t) \in [K]$. The action vector of all players at time $t$ is denoted as $S(t) := [s_1(t), ..., s_M(t)]$, which is referred to as a "matching" for convenience although it is not necessarily one-to-one.

**Individual Outcomes.** For each player $m$, an outcome[1] $O_{k,m}(t)$ is associated with her action of pulling arm $s_m(t) = k$ at time $t$, which is defined as

$$O_{k,m}(t) := X_{k,m}(t)\eta_k(S(t)). \tag{1}$$

In Eqn. (1), $X_{k,m}(t)$ is a random variable of arm utility and $\eta_k(S(t))$ is the no-collision indicator defined by $\eta_k(S) := \mathbb{1}\{|C_k(S)| \leq 1\}$ with $C_k(S) := \{n \in [M]|s_n = k\}$. In other words, if player $m$ is the only player choosing arm $k$, the outcome is $X_{k,m}(t)$; if multiple players choose arm $k$ simultaneously, a collision happens on this arm and the outcome is zero regardless of $X_{k,m}(t)$.

For a certain arm-player pair, i.e., $(k, m)$, the set of random arm utilities $\{X_{k,m}(t)\}_{t \geq 1}$ is assumed to be sampled independently from an unknown distribution $\phi_{k,m}$, which has a bounded support on $[0, 1]$ and an unknown expectation $\mathbb{E}[X_{k,m}(t)] = \mu_{k,m}$. In general, these utility distributions are player-dependent, i.e., $\mu_{k,m} \neq \mu_{k,n}$ when $m \neq n$. Note that despite the time independence among $\{X_{k,m}(t)\}_{t \geq 1}$ for a certain arm-player pair $(k, m)$, correlations can exist among the random utility variables of different arm-player pairs, i.e., among $X_{k,m}(t)$ for different $(k, m)$ pairs.

To ease the exposition, we define $\mathcal{S} = \{S = [s_1, ..., s_M]|s_m \in [K], \forall m \in [M]\}$ as the set of all possible matchings $S$ and abbreviate the arm $k$ of player $m$ as arm $(k, m)$. We further denote $\boldsymbol{\mu} = [\mu_{k,m}]_{(k,m) \in [K] \times [M]}$ and $\boldsymbol{\mu}_S = [\mu_{s_m,m}]_{m \in [M]}$ for $S = [s_1, ..., s_M]$.

**System Rewards.** Besides players' individual outcomes, with matching $S(t)$ chosen at time $t$, a random system reward, denoted as $V(S(t), t)$, is collected for the entire system. The most commonly-

---

[1]The term "outcome" distinguishes players' individual rewards from the later introduced system rewards.

studied reward function (Bistritz and Leshem, 2020; Boursier et al., 2020) is the sum of outcomes from different players (referrd to as the *linear reward function*), i.e., $V(S(t), t) := \sum_{m \in [M]} O_{s_m(t), m}(t)$. With this linear reward function, for matching $S$, the expected system reward is denoted as $V_{\boldsymbol{\mu}, S} := \mathbb{E}[V(S, t)] = \sum_{m \in [M]} \mu_{s_m, m} \eta_{s_m}(S)$ under matrix $\boldsymbol{\mu}$. As almost all of the existing MP-MAB literature focus on the linear reward function, we also focus on this case first, but note that the problem formulation presented in this section can be extended to general (nonlinear) reward functions in Section 5.

**Feedback Model.** Different feedback models exist in the MP-MAB literature, and this work focuses on the collision-sensing model (Bistritz and Leshem, 2018, 2020; Boursier and Perchet, 2019; Boursier et al., 2020). Specifically, player $m$ can access her own outcome $O_{s_m(t), m}(t)$ and the corresponding no-collision indicator $\eta_{s_m(t)}(S(t))$, but neither the overall reward $V(S(t), t)$ nor outcomes of other players. In other words, at time $t$, player $m$ chooses arm $s_m(t)$ based on her own history $H_m(t) = \left\{ s_m(\tau), O_{s_m(\tau), m}(\tau), \eta_{s_m(\tau)}(S(\tau)) \right\}_{1 \leq \tau \leq t-1}$.

**Regret Definition.** If $\boldsymbol{\mu}$ is known *a priori*, the optimal choice is the matching that gives the highest expected reward $V_{\boldsymbol{\mu}, *} := \max_{S \in \mathcal{S}} V_{\boldsymbol{\mu}, S}$. We formally define the regret after $T$ rounds of playing as

$$R(T) = TV_{\boldsymbol{\mu}, *} - \mathbb{E}\left[ \sum_{t=1}^{T} V(S(t), t) \right], \tag{2}$$

where the expectation is w.r.t. the randomness of the policy and the environment.

One technical novelty worth noting is that this work considers the general case with possibly *multiple optimal matchings*, instead of the commonly assumed unique one (Bistritz and Leshem, 2018, 2020). Multiple optimal matchings might be uncommon for the linear reward function, but often occur under more sophisticated reward functions that will be discussed later, e.g., the minimal function, and brings substantial difficulties into player coordination. In addition, the proposed BEACON design is also applicable to the homogeneous setting, i.e., $\forall m \in [M], \mu_{k,m} = \mu_k$, with some trivial adjustments.

## 3 The BEACON Algorithm

### 3.1 Algorithm Structure and Key Ideas

After the orthogonalization procedure (Wang et al., 2020) at the beginning of the game, during which each player individually estimates the number of players $M$ and assigns herself of a unique index $m \in [M]$, BEACON proceeds in epochs and each epoch consists of two phases: (implicit) communication and exploration.[2] While similar two-phase structures have been adopted by other heterogeneous MP-MAB algorithms (Tibrewal et al., 2019; Boursier et al., 2020), those designs fail to have regrets approaching the centralized lower bound.

The challenge in approaching the centralized lower bound is not only designing more efficient implicit communications and explorations, but also connecting them in a way that neither phase dominates the overall regret and both approach the centralized lower bound simultaneously. BEACON precisely achieves these goals, with several key ideas that not only are crucial to closing the regret gap but also hold individual values in MP-MAB research. First, a novel adaptive differential communication (ADC) method is proposed, which is fundamental in improving the effectiveness and efficiency of implicit communications. Specifically, ADC drastically reduces the communication cost from up to $O(\log(T))$ per epoch in state-of-the-art designs (Boursier et al., 2020) to $O(1)$ per epoch, which ensures a low communication cost. Second, CUCB principles (Chen et al., 2013) are incorporated with a batched exploration structure to ensure a low exploration loss (see Section 8 for more discussions on the relationship between CMAB and MP-MAB). CUCB principles address a critical challenge of *large amount of matchings* in heterogeneous MP-MAB (i.e., $|\mathcal{S}| = K^M$), which hampered prior designs. The batched structure, on the other hand, is carefully embedded and optimized such that the need of communication and exploration is balanced, leading to neither dominating the overall regret.

---

[2]Details of the orthogonalization procedure are given in the supplementary material. In addition, by "exploration phase", we mean the time steps in one epoch that are not used for (implicit) communications, which actually contain both exploration and exploitation.

## 3.2 Batched Exploration

To facilitate the illustration, we first present the batched exploration scheme and also a sketch of BEACON under an imaginary communication-enabled setting, which will be addressed in Section 3.3. Specifically, players are assumed to be able to communicate with each other freely in this subsection.

The batched exploration proceeds as follows. At the beginning of epoch $r$, each player $m$ maintains an arm counters $p_{k,m}^r$ for each arm $k$ of hers. The counters are updated as $p_{k,m}^r = \lfloor \log_2(T_{k,m}^r) \rfloor$, where $T_{k,m}^r$ is the number of exploration pulls on arm $(k,m)$ up to epoch $r$. Then, the leader (referring to the player with index 1) collects arm statistics from followers (referring to the players other than the leader). Specifically, if $p_{k,m}^r > p_{k,m}^{r-1}$, statistics $\tilde{\mu}_{k,m}^r$ is collected from follower $m$; otherwise, $\tilde{\mu}_{k,m}^r$ is not updated and kept the same as $\tilde{\mu}_{k,m}^{r-1}$, where $\tilde{\mu}_{k,m}^r$ is a to-be-specified characterization of arm $(k,m)$'s sample mean $\hat{\mu}_{k,m}^r$. With the updated information, an upper confidence bound (UCB) matrix $\bar{\boldsymbol{\mu}}_r = [\bar{\mu}_{k,m}^r]_{(k,m)\in[K]\times[M]}$ is calculated by the leader, where $\bar{\mu}_{k,m}^r = \tilde{\mu}_{k,m}^r + \sqrt{3 \ln t_r / 2^{p_{k,m}^r + 1}}$, and $t_r$ is the time step at the beginning of epoch $r$.

The UCB matrix $\bar{\boldsymbol{\mu}}_r$ is then fed into a combinatorial optimization solver, denoted as $\texttt{Oracle}(\cdot)$, which outputs the optimal matching w.r.t. the input. Specifically, $S_r = [s_1^r, ..., s_M^r] \leftarrow \texttt{Oracle}(\bar{\boldsymbol{\mu}}_r) = \arg\max_{S\in\mathcal{S}} \left\{ \sum_{s_m} \bar{\mu}_{s_m,m}^r \right\}$, which can be computed with a polynomial time complexity using the Hungarian algorithm (Munkres, 1957). We note that similar optimization solvers are also required by Boursier et al. (2020); Tibrewal et al. (2019). Inspired by the exploration choice of CUCB, this matching $S_r$ is chosen to be explored. The leader thus assigns the matching $S_r$ to followers, i.e., arm $s_m^r$ for player $m$.

After the assignment, the exploration begins. One important ingredient of BEACON is that the duration of exploring the chosen matching, i.e., the adopted batch size, is determined by the smallest arm counter in it. Specifically, for $S_r$, we denote $p_r = \min_{m\in[M]} p_{s_m^r,m}^r$ and the batch size is chosen to be $2^{p_r}$. In other words, during the following $2^{p_r}$ time steps, players are fixated to exploring the matching $S_r$. Then, epoch $r+1$ starts, and the same procedures are iterated.

**Remarks.** BEACON directly selects the matching with the largest UCB to explore. It turns out that this natural method significantly outperforms the "matching-elimination" scheme in Boursier et al. (2020), and is critical to achieving a near-optimal exploration loss. In addition, the chosen batch size of $2^{p_r}$ ensures *sufficient but not excessive* pulls w.r.t. the least pulled arm(s) in the chosen matching, which dominate the uncertainties. Furthermore, while similar batched structures have been utilized in the bandit literature (Auer et al., 2002; Hillel et al., 2013), the updating of arm counters in BEACON is carefully tailored. Last, the leader collects followers' statistics only when arm counters increase, i.e., $p_{k,m}^r > p_{k,m}^{r-1}$, which means $\tilde{\mu}_{k,m}^r$ is sufficiently more precise than $\tilde{\mu}_{k,m}^{r-1}$. This design contributes to a low communication frequency while not affecting the exploration efficiency.

## 3.3 Efficient Implicit Communication

Since explicit communication is prohibited in decentralized MP-MAB problems, we now discuss how to use *implicit* communication (Boursier and Perchet, 2019) to share information in BEACON. Specifically, players can take predetermined turns to "communicate" by having the "receive" player sample one arm and the "send" player either pull (create collision; bit 1) or not pull (create no collision; bit 0) the same arm to transmit one-bit information. Although information sharing is enabled, such a forced-collision communication approach is inevitably costly, as collisions reduce the rewards. The challenge now is how to keep the communication loss small, ideally $O(\log(T))$.

The *batched* exploration scheme plays a key role in reducing the communication loss via infrequent information updating. In other words, players only communicate statistics and decisions before each batch instead of each time step. With the aforementioned batch size, there are at most $O(\log(T))$ epochs in horizon $T$. Thus, intuitively, if the communication loss per epoch can be controlled of order $O(1)$ irrelevant of $T$, the overall communication loss would not be dominating. However, this requirement is challenging and none of the existing implicit communication schemes (Boursier and Perchet, 2019; Boursier et al., 2020) can meet it, which calls for a novel communication design.

From the discussion of the exploration phases, we can see that sharing arm statistics $\tilde{\mu}_{k,m}^r$ is the most challenging part. Specifically, as opposed to sharing integers of arm indices in $S_r$ and the batch

size parameter $p_r$, statistics $\tilde{\mu}_{k,m}^r$ is often a decimal while forced-collision is fundamentally a digital communication protocol. We thus focus on the communication design for sharing statistics $\tilde{\mu}_{k,m}^r$, and propose the adaptive differential communication (ADC) method as detailed below. Details of sharing $S_r$ and $p_r$ can be found in supplementary material.

The first important idea is to let followers **adaptively** quantize sample means for communication. Specifically, upon communication, the arm statistics $\tilde{\mu}_{k,m}^r$ is not directly set as the collected sample mean $\hat{\mu}_{k,m}^r$. Instead, $\tilde{\mu}_{k,m}^r$ is a quantized version of $\hat{\mu}_{k,m}^r$ using $\lceil 1 + p_{k,m}^r/2 \rceil$ bits. Since $\tilde{\mu}_{k,m}^r$ is communicated only upon an increase of the arm counter $p_{k,m}^r$, this quantization length is adaptive to the arm counter (or equivalently the arm pulls), and further to the adopted confidence bound in Section 3.2, i.e., $\sqrt{3 \ln t_r / 2^{p_{k,m}^r+1}}$. However, this idea alone is not sufficient because $p_{k,m}^r$ is of order up to $O(\log(T))$, instead of $O(1)$.

To overcome this obstacle, the second key idea is **differential** communication, which significantly reduces the redundancies in statistics sharing. Specifically, follower $m$ first computes the difference $\tilde{\delta}_{k,m}^r = \tilde{\mu}_{k,m}^r - \tilde{\mu}_{k,m}^{r-1}$, and then truncates the bit string of $\tilde{\delta}_{k,m}^r$ upon the most significant non-zero bit, e.g., 110 for 000110. She only communicates this truncated version of $\tilde{\delta}_{k,m}^r$ in the transmission of $\tilde{\mu}_{k,m}^r$ to the leader. The intuition is that $\tilde{\mu}_{k,m}^r$ and $\tilde{\mu}_{k,m}^{r-1}$ are both concentrated at $\mu_{k,m}$ with high probabilities, which results in a small $\tilde{\delta}_{k,m}^r$. From an information-theoretic perspective, the conditional entropy of

---

**Algorithm 1** BEACON: Leader

1: Initialization: $r \leftarrow 0$; $\forall(k,m), p_{k,m}^r \leftarrow -1, T_{k,m}^r \leftarrow 0, \tilde{\mu}_{k,m}^r \leftarrow 0$
2: Play each arm $k \in [K]$ and $T_{k,1}^{r+1} \leftarrow T_{k,1}^r + 1$
3: **while** not reaching the time horizon **do**
4:    $r \leftarrow r + 1$
5:    $\forall(k,m), p_{k,m}^r \leftarrow \lfloor \log_2(T_{k,m}^r) \rfloor$
6:    $\forall k \in [K]$, update sample mean $\hat{\mu}_{k,1}^r$ with the first $2^{p_{k,1}^r}$ exploratory samples from arm $k$
   ▷ *Communication Phase*
7:    **for** $(k,m) \in [K] \times [M]$ **do**
8:       **if** $p_{k,m}^r > p_{k,m}^{r-1}$ **then**
9:          $\tilde{\delta}_{k,m}^r \leftarrow \texttt{Receive}(\tilde{\delta}_{k,m}^r, m)$
10:         $\tilde{\mu}_{k,m}^r \leftarrow \tilde{\mu}_{k,m}^{r-1} + \tilde{\delta}_{k,m}^r$
11:       **else**
12:         $\tilde{\mu}_{k,m}^r \leftarrow \tilde{\mu}_{k,m}^{r-1}$
13:       **end if**
14:    **end for**
15:    $\forall(k,m), \bar{\mu}_{k,m}^r \leftarrow \tilde{\mu}_{k,m}^r + \sqrt{3 \ln t_r / 2^{p_{k,m}^r+1}}$
16:    $S_r = [s_1^r, ..., s_M^r] \leftarrow \texttt{Oracle}(\bar{\mu}_r)$
17:    $\forall m \in [M], \texttt{Send}(s_m^r, m)$
   ▷ *Exploration Phase*
18:    $p_r \leftarrow \min_{m \in [M]} p_{s_m^r, m}^r$
19:    Play arm $s_1^r$ for $2^{p_r}$ times
20:    Signal followers to stop exploration
21:    Update $\forall m \in [M], T_{s_m,m}^{r+1} \leftarrow T_{s_m,m}^r + 2^{p_r}$
22: **end while**

---

$\tilde{\mu}_{k,m}^r$ on $\tilde{\mu}_{k,m}^{r-1}$, i.e., $H(\tilde{\mu}_{k,m}^r | \tilde{\mu}_{k,m}^{r-1})$, is often small because they are highly correlated.[3]

As will be clear in the regret analysis, putting these two ideas together results in an effective communication design, i.e, the ADC scheme, whose expected regret is of order $O(1)$ per epoch and $O(\log(T))$ overall. This method itself represents an important improvement over prior implicit communication protocols in MP-MAB, whose loss is typically of order $O(\log(T))$ per epoch and $O(\log^2(T))$ in total with multiple optimal matchings (Boursier and Perchet, 2019; Boursier et al., 2020). Techniques similar to ADC have been utilized in areas outside of MAB, e.g., wireless communications (Goldsmith and Chua, 1998), with proven success in practice (Goldsmith, 2005).

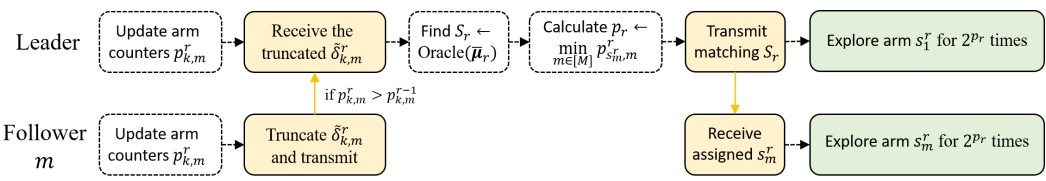

Figure 1: A sketch of epoch $r$ in BEACON. Yellow boxes and yellow lines indicate communications, green boxes for explorations, and boxes with dotted frame for computations.

---

[3]Note that sharing the truncated version of $\tilde{\delta}_{k,m}^r$ results in another difficulty that its length varies for different player-arm pairs and is unknown to the leader. A specially crafted "signal-then-communicate" scheme is designed to tackle this challenge and can be found in the supplementary material.

The complete BEACON algorithm can now be obtained by plugging ADC into the batched exploration structure. A sketch of one BEACON epoch is illustrated in Fig. 1, and the leader's algorithm is presented in Algorithm 1. The follower's algorithm can be found in the supplementary material, along with the definitions of the implicit communication protocols denoted by functions `Send()` and `Receive()`. Note that the for-loops with $(k, m)$ and $\forall (k, m)$ in the pseudo-codes indicate the iteration over all possible arm-player pairs of $[K] \times [M]$. In addition, the communications of the leader to herself indicated by the pseudo-codes denote her own calculations instead of real forced-collision communications (among the leader and followers), which is a simplification for better exposition.

## 4 Theoretical Analysis

With notations $\mathcal{S}_c := \{S \in \mathcal{S} | \exists m \neq n, s_m = s_n\}$ as the set of collided matchings; $\mathcal{S}_* := \{S \in \mathcal{S} | V_{\boldsymbol{\mu}, S} = V_{\boldsymbol{\mu}, *}\}$ as the set of optimal matchings; $\mathcal{S}_b = \mathcal{S} \backslash (\mathcal{S}_* \cup \mathcal{S}_c)$ as the set of collision-free suboptimal matchings; $\Delta_{\min}^{k,m} := V_{\boldsymbol{\mu}, *} - \max\{V_{\boldsymbol{\mu}, S} | S \in \mathcal{S}_b, s_m = k\}$ as the minimum sub-optimality gap for collision-free matchings containing arm-player pair $(k, m)$; $\Delta_{\min} := \min_{(k,m)}\{\Delta_{\min}^{k,m}\}$ as the minimum sub-optimality gap for all collision-free matchings, the regret of BEACON with the linear reward function is analyzed in the following theorem.

**Theorem 1.** *With the linear reward function, the regret of BEACON is upper bounded as*[4]

$$
R_{\text{linear}}(T) = \tilde{O} \left( \sum_{(k,m) \in [K] \times [M]} \frac{M \log(T)}{\Delta_{\min}^{k,m}} + M^2 K \log(T) \right) \tag{3}
$$
$$
= \tilde{O} \left( \frac{M^2 K \log(T)}{\Delta_{\min}} \right).
$$

Note that in Eqn. (3), the first term represents the exploration regret of BEACON, and the second term the communication regret. Compared with the state-of-the-art regret result $\tilde{O}(M^3 K \log(T)/\Delta_{\min})$ for METC (Boursier et al., 2020), the regret bound in Theorem 1 improves the dependence of $M$ from $M^3$ to $M^2$. It turns out that this quadratic dependence is optimal because the same dependence exists in the centralized lower bound (hence a natural lower bound for decentralized MP-MAB) for the linear reward function, as from Kveton et al. (2015c):[5]

$$
R_{\text{linear}}(T) = \Omega \left( \frac{M^2 K}{\Delta_{\min}} \log(T) \right). \tag{4}
$$

By comparing Theorem 1 and Eqn. (4), it can be observed that with the linear reward function, BEA-CON achieves a regret that approaches the centralized lower bound. The efficiency and effectiveness of both exploration and communication phases are critical in this achievement, as we can see that both terms in Theorem 1 are non-dominating at $\tilde{O}(M^2 K \log(T))$.

In addition to the problem-dependent bound given in Theorem 1, the following theorem establishes a problem-independent bound, which can be thought of as a worst-case characterization.

**Theorem 2.** *With the linear reward function, it holds that*

$$
R_{\text{linear}}(T) = O\left(M\sqrt{KT \log(T)}\right).
$$

Theorem 2 not only improves the best known problem-independent bound $O(M^{\frac{3}{2}}\sqrt{KT \log(T)})$ (Boursier et al., 2020) in the decentralized MP-MAB literature, but also approaches the centralized lower bound $\Omega(M\sqrt{KT})$ (Kveton et al., 2015c; Merlis and Mannor, 2020) up to logarithmic factors.

Theorems 1 and 2 demonstrate that for the linear reward function, BEACON closes the performance gap (both problem-dependent and problem-independent) between decentralized heterogeneous MP-MAB algorithms and their centralized counterparts. The regret bounds of various MP-MAB algorithms, including BEACON, are summarized in Table 1.

---

[4]With the notation $\tilde{O}(\cdot)$, logarithmic parameters containing $K$ are ignored.

[5]This lower bound holds for the cases with arbitrarily correlated arms, as considered in this work. Under additional arm independence assumptions (Combes et al., 2015), lower regrets can be achieved.

**Remarks.** We note that it is also feasible to combine the ADC protocol and METC (Boursier et al., 2020), which can address its communication inefficiency, especially with multiple optimal matchings. However, with ideas from CUCB, BEACON is much more efficient in exploration than "Explore-then-Commit"-type of algorithms (e.g., METC), which is the main reason we did not fully elaborate the combination of METC and ADC in this work. Theoretically, this superiority can be reflected in the extra multiplicative factor in the exploration loss of METC shown in Table 1.

Table 1: Regret Bounds of Decentralized MP-MAB Algorithms

| Algorithm/Reference | Reward function | Assumptions | | | Regret |
| --- | --- | --- | --- | --- | --- |
| | | Known horizon $T$ | Known gap $\Delta_{\min}$ | Unique optimal matching | |
| GoT † (Bistritz and Leshem, 2020) | Linear | No | Yes | Yes | $O\left(M\log^{1+\kappa}(T)\right)$ |
| Decentralized MUMAB (Magesh and Veeravalli, 2019) | Linear | No | Yes | No | $O\left(K^3\log(T)\right)$ |
| ESE1 (Tibrewal et al., 2019) | Linear | No | No | Yes | $O\left(\frac{M^2K}{\Delta_{\min}^2}\log(T)\right)$ |
| METC (Boursier et al., 2020) | Linear | Yes | No | Yes | $O\left(\frac{M^3K}{\Delta_{\min}}\log(T)\right)$ |
| METC (Boursier et al., 2020) | Linear | Yes | No | No | $O\left(MK\left(\frac{M^2\log(T)}{\Delta_{\min}}\right)^{1+\iota}\right)$ |
| BEACON (this work, Thm. 3) | General | No | No | No | $\tilde{O}\left(\frac{MK\Delta_{\max}}{(f^{-1}(\Delta_{\min}))^2}\log(T)\right)$ |
| BEACON (this work, Thm. 1) | Linear | No | No | No | $\tilde{O}\left(\frac{M^2K}{\Delta_{\min}}\log(T)\right)$ |
| Lower bound (Kveton et al., 2015c) | Linear | N/A | N/A | N/A | $\Omega\left(\frac{M^2K}{\Delta_{\min}}\log(T)\right)$ |

†: tuning parameters in GoT requires knowledge of arm utilities;
$\kappa, \iota$: arbitrarily small non-zero constants.

## 5 Beyond Linear Reward Functions

### 5.1 General Reward Functions

In this section, we move away from the linear reward functions in almost all prior MP-MAB research, and extend the study to general (nonlinear) reward functions. Two exemplary nonlinear reward functions are given below, with more examples provided in the supplementary material.

- Proportional fairness: $V(S,t) = \sum_{m\in[M]} \omega_m \ln(\epsilon + O_{s_m,m}(t))$, where $\epsilon > 0$ and $\omega_m > 0$ are constants. It promotes fairness among players (Mo and Walrand, 2000);
- Minimal: $V(S,t) = \min_{m\in[M]}\{O_{s_m,m}(t)\}$, which indicates the system reward is determined by the least-rewarded player, i.e., the short board of the system;[6]

These reward functions all hold their value in real-world applications, but are largely ignored and cannot be effectively solved by previous approaches. The difficulty introduced by this extension not only lies in the complex mapping from the (unreliable) individual outcomes to system rewards, but also comes from the potential "coupling" effect among players (e.g., the minimal reward function).

To better characterize the problem, the following mild assumptions are considered.

**Assumption 1.** *There exists an expected reward function $v(\cdot)$ such that $V_{\boldsymbol{\mu},S} := \mathbb{E}[V(S,t)] = v(\boldsymbol{\mu}_S \odot \boldsymbol{\eta}_S)$, where $\boldsymbol{\eta}_S := [\eta_{s_m}(S)]_{m\in[M]}$ and $\boldsymbol{\mu}_S \odot \boldsymbol{\eta}_S := [\mu_{s_m,m}\eta_{s_m}(S)]_{m\in[M]}$.*

**Assumption 2** (Monotonicity)**.** *The expected reward function is monotonically non-decreasing with respect to the vector $\boldsymbol{\Lambda} = \boldsymbol{\mu}_S \odot \boldsymbol{\eta}_S$, i.e., if $\boldsymbol{\Lambda} \preceq \boldsymbol{\Lambda}'$, we have $v(\boldsymbol{\Lambda}) \leq v(\boldsymbol{\Lambda}')$.*

**Assumption 3** (Bounded smoothness)**.** *There exists a strictly increasing (and thus invertible) function $f(\cdot)$ such that $\forall \boldsymbol{\Lambda}, \boldsymbol{\Lambda}', |v(\boldsymbol{\Lambda}) - v(\boldsymbol{\Lambda}')| \leq f(\|\boldsymbol{\Lambda} - \boldsymbol{\Lambda}'\|_\infty)$.*

Assumption 1 indicates that the expected reward $V_{\boldsymbol{\mu},S}$ of matching $S$ is determined only by its expected individual outcomes. It is true for the linear reward function, and also generally holds if

---

[6]Differences with the max-min fairness (Bistritz et al., 2020) are elaborated in the supplementary material.

distributions $\{\phi_{k,m}\}$ are mutually independent and determined by their expectations $\{\mu_{k,m}\}$, e.g., Bernoulli distribution. Assumptions 2 and 3 concern the monotonicity and smoothness of the expected reward function, which are natural for most practical reward functions, including the above examples. Similar assumptions have been adopted by Chen et al. (2013, 2016b); Wang and Chen (2018).

## 5.2 BEACON Adaption and Performance Analysis

In Section 3.2, a combinatorial optimization solver $\mathtt{Oracle}(\cdot)$ is implemented for the linear reward function. With ideas from CUCB (Chen et al., 2013), BEACON can be extended to handle a general reward function with a corresponding solver $\mathtt{Oracle}(\cdot)$ that outputs the optimal (non-collision) matching w.r.t. the input matrix $\boldsymbol{\mu}'$, i.e., $S' \leftarrow \mathtt{Oracle}(\boldsymbol{\mu}') = \arg\max_{S \in \mathcal{S} \setminus \mathcal{S}_c} V_{\boldsymbol{\mu}',S}$.

With such an oracle, the following theorem provides performance guarantees of BEACON.

**Theorem 3** (**General reward function**). *Under Assumptions 1, 2, and 3, denoting* $\Delta_{\max}^{k,m} := V_{\boldsymbol{\mu},*} - \min\{V_{\boldsymbol{\mu},S} | S \in \mathcal{S}_b, s_m = k\}$ *and* $\Delta_c := f(1)$, *the regret of BEACON is upper bounded as*

$$
R(T) = \tilde{O} \left( \sum_{(k,m) \in [K] \times [M]} \left[ \frac{\Delta_{\min}^{k,m}}{(f^{-1}(\Delta_{\min}^{k,m}))^2} + \int_{\Delta_{\min}^{k,m}}^{\Delta_{\max}^{k,m}} \frac{1}{(f^{-1}(x))^2} \mathrm{d}x \right] \log(T) + M^2 K \Delta_c \log(T) \right)
$$

$$
= \tilde{O} \left( \sum_{(k,m) \in [K] \times [M]} \frac{\Delta_{\max}^{k,m} \log(T)}{(f^{-1}(\Delta_{\min}^{k,m}))^2} + M^2 K \Delta_c \log(T) \right).
$$

With a stronger smoothness assumption, we can obtain a clearer exposition of the regret.

**Corollary 1.** *Under Assumptions 1 and 2, if there exists* $B > 0$ *such that* $\forall \boldsymbol{\Lambda}, \boldsymbol{\Lambda}', |v(\boldsymbol{\Lambda}) - v(\boldsymbol{\Lambda}')| \leq B \|\boldsymbol{\Lambda} - \boldsymbol{\Lambda}'\|_\infty$, *it holds that*

$$
R(T) = \tilde{O} \left( \sum_{(k,m) \in [K] \times [M]} \frac{B^2}{\Delta_{\min}^{k,m}} \log(T) + M^2 K B \log(T) \right).
$$

In addition, since the combinatorial optimization problems with general reward functions can be NP-hard, it is more practical to adopt approximate solvers rather than the exact ones (Vazirani, 2013). To accommodate such needs, we introduce the following definition of $(\alpha, \beta)$-approximation oracle for $\alpha, \beta \in [0, 1]$ as in Chen et al. (2013, 2016a,b); Wang and Chen (2017):

**Definition 1.** *With a matrix* $\boldsymbol{\mu}' = [\mu'_{k,m}]_{(k,m) \in [K] \times [M]}$ *as input, an* $(\alpha, \beta)$-*approximation oracle outputs a matching* $S'$, *such that* $\mathbb{P}[V_{\boldsymbol{\mu}',S'} \geq \alpha \cdot V_{\boldsymbol{\mu}',*}] \geq \beta$, *where* $V_{\boldsymbol{\mu}',*} = \max_{S \in \mathcal{S}} V_{\boldsymbol{\mu}',S}$.

With only an approximate solver, it is no longer fair to compare the performance against the optimal reward. Instead, as in the CMAB literature (Chen et al., 2013, 2016a,b; Wang and Chen, 2017), an $(\alpha, \beta)$-approximation regret is considered: $R_{\alpha,\beta}(T) = T\alpha\beta V_{\boldsymbol{\mu},*} - \mathbb{E}[\sum_{t=1}^{T} V(S(t), t)]$, where the performance is compared to the $\alpha\beta$ fraction of the optimal reward. As shown in the supplementary material, for this $(\alpha, \beta)$-approximation regret, an upper bound similar to Theorem 3 can be obtained.

# 6 Experiments

In this section, BEACON is empirically evaluated with both linear and general (nonlinear) reward functions. All results are averaged over 100 experiments and the utilities follow mutually independent Bernoulli distributions. Additional experimental details, empirical algorithm enhancements and more experimental results (e.g., with a large game), can be found in the supplementary material.

**Linear Reward Function.** BEACON is evaluated along with the centralized CUCB (Chen et al., 2013) and the state-of-the-art decentralized algorithm METC (Boursier et al., 2020). The decentralized GoT algorithm (Bistritz and Leshem, 2020) is also evaluated but its regrets are over $100\times$ larger than those of BEACON, and thus is omitted in the plots. Fig. 2(a) reports results under the same instance in Boursier et al. (2020) with $K = 5, M = 5$. Although this is a relatively hard instance with multiple optimal matchings and small sub-optimality gaps, BEACON still achieves a comparable performance as CUCB, and significantly outperforms METC: an approximate $7\times$ regret reduction at the horizon.

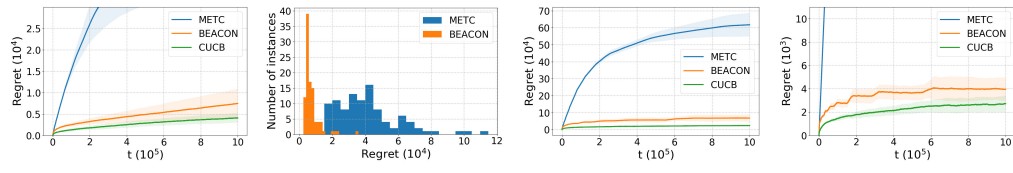

| (a) Linear, cumul. regret. | (b) Linear, regret histo. | (c) Proportional fairness. | (d) Minimal. |

Figure 2: Regret comparisons. The continuous curves represent the empirical average values, and the shadowed areas represent the standard deviations. (a), (c) and (d) are evaluated with specific game instances, and (b) is the regret histogram of 100 randomly generated instances.

To validate whether this significant gain of BEACON over METC is representative, we plot in Fig. 2(b) the histogram of regrets with 100 randomly generated instances still with $M = 5, K = 5, T = 10^6$. Expected arm utilities are uniformly sampled from $[0, 1]$ in each instance. It can be observed that the gain of BEACON is very robust – its average regret is approximately $6\times$ lower than METC.

**General Reward Function.** Two representative nonlinear reward functions are used to evaluate BEACON: (1) the proportional fairness function with $\forall m \in [M], \omega_m = 1, \epsilon = 10^{-2}$; (2) the minimal function. BEACON is compared with CUCB and METC.[7] Under a game instance with $M = 6, K = 8$, Fig. 2(c) reports the regrets under the proportional fairness function, and Fig. 2(d) with the minimal function. From both results, it can be observed that BEACON has slightly larger (but comparable) regrets than the centralized CUCB, while significantly outperforming METC.

To summarize, BEACON not only significantly outperforms state-of-the-art decentralized MP-MAB algorithms, but is also capable of *empirically* approaching the centralized performance, which is the first time for a decentralized heterogeneous MP-MAB algorithm to the best of our knowledge.

## 7  Discussions

We briefly summarize the novel theoretical contributions of this work:

- **Closing the regret gap.** With the linear reward function, BEACON can approach (both problem-dependent and problem-independent) centralized lower bounds. To the best of our knowledge, this is the first time such performance gap is closed (scaling wise) for the heterogeneous MP-MAB.
- **Broader applicability.** BEACON can handle a broad range of general reward functions with a regret of $O(\log(T))$, while existing algorithms mostly focus on the linear reward function and their analyses do not apply to the general reward functions. To the best of our knowledge, this is the first time general reward functions are studied in decentralized MP-MAB.
- **Fewer assumptions.** BEACON achieves a strictly $O(\log(T))$-regret without any assumptions or prior knowledge of the game instance, while prior MP-MAB algorithms typically rely on additional assumptions or knowledge; see Table 1 for details.

In addition to these tangible contributions, this work also demonstrates the benefit of incorporating CMAB techniques in the study of MP-MAB. In this paper, both the BEACON design and its regret analysis benefit from CMAB, especially CUCB (Chen et al., 2013; Kveton et al., 2015c). While these two sub-fields of MAB are largely considered disjoint, this work shows that the underlying connection is rather fundamental. This revelation may open up interesting future research directions. For example, under the structure of BEACON, it is conceivable to introduce more advanced CMAB algorithms, e.g., ESCB (Combes et al., 2015), into the study of MP-MAB with additional assumptions on the arm dependence. In another direction, ideas from this work may also contribute to the study of CMAB. For example, due to the batched structure, BEACON only accesses the oracle $O(\log(T))$ times over $T$ steps, which is more computational efficient than the $O(T)$ times access in CUCB.

Besides contributions, there are open questions left for future studies. First, BEACON relies on a centralized combinatorial optimization solver, i.e., Oracle$(\cdot)$, and so do Boursier et al. (2020); Tibrewal et al. (2019). While being a reasonable requirement, this oracle might be computational-infeasible for some applications, e.g., with Internet-of-Things (IoT) devices, especially when $M$ and $K$ are large. Also, while the oracle allows general analysis, it also decouples the problem into

---

[7]To make meaningful comparisons, non-trivial adjustments and enhancements have been applied to METC, which originally applies only to the linear reward function. Details are given in the supplementary material.

two disconnected parts: combinatorial optimization and bandits. It might be helpful to tailor the algorithm into one specific reward function, where joint designs over these two parts can be performed. Furthermore, it would be interesting to investigate the non-cooperative setting as in Boursier and Perchet (2020), where we believe the design ideas in this work can still be of use, especially ADC.

## 8 Related Works

**Decentralized MP-MAB.** Since Liu and Zhao (2010), most MP-MAB works consider the homogeneous variant with player-independent arm utilities (Avner and Mannor, 2014; Rosenski et al., 2016; Besson and Kaufmann, 2018). With implicit communications, Boursier and Perchet (2019); Wang et al. (2020) prove regrets that approach the centralized ones. The homogeneous variant is fairly well understood by now. The heterogeneous MP-MAB problems (Kalathil et al., 2014; Nayyar et al., 2016) with player-dependent arm utilities, on the other hand, remain largely open. The recent attempts have been summarized in Table 1, whose regrets are far from the (natural) centralized lower bound. Note that a similar idea of adaptive quantization is applied by Boursier et al. (2020), but the differential communication part in ADC is entirely novel and more critical to the overall performance.

All the aforementioned works are confined to the linear reward model. To the best of our knowledge, this work is the first to study general reward functions. Fairness is considered in Bistritz et al. (2020) but with major differences elaborated in the supplementary material. Other MP-MAB variants, including "stable" allocations (Avner and Mannor, 2016; Darak and Hanawal, 2019), no-sensing (Lugosi and Mehrabian, 2018; Shi et al., 2020; Bubeck and Budzinski, 2020; Bubeck et al., 2021), and adversarial (Alatur et al., 2020; Bubeck et al., 2020; Shi and Shen, 2021), fall out of our scope.

**Combinatorial MAB.** Since first presented by Chen et al. (2013), many variants of stochastic CMAB have been investigated (Kveton et al., 2014, 2015a). Some recent works have also introduced Thompson Sampling into CMAB (Wang and Chen, 2018; Perrault et al., 2020). The study of lower bounds in CMAB has been active, e.g., for the linear reward function with correlated arms (Kveton et al., 2015b; Degenne and Perchet, 2016) and independent arms (Combes et al., 2015). Recent attempts on lower bounds for general reward functions are reported by Merlis and Mannor (2020).

As illustrated in the design of BEACON, the decentralized MP-MAB model is closely related to CMAB, while these connections are largely ignored in the previous works. With more details presented in the supplementary material, we here briefly note that in some sense, MP-MAB can be thought of as a decentralized version of CMAB, and this decentralized nature leads to additional challenges with collision-avoidance and information sharing.

## 9 Conclusion

In this work, we first investigated decentralized heterogeneous MP-MAB problems with linear reward function and proposed the BEACON algorithm. A novel adaptive differential (implicit) communication approach was designed and a batched structure was carefully crafted to incorporate the exploration principles from CUCB. With these novel ideas, BEACON achieved regrets that not only improve all prior regret bounds but in fact approach the centralized lower bound for the first time in the study of decentralized heterogeneous MP-MAB. Then, we extended the study to general reward functions and showed that BEACON can still obtain a regret of $O(\log(T))$ with simple modifications. Experimental results demonstrated that the gain of BEACON does not exist just in the theoretical analysis – significant gains over state-of-the-art decentralized algorithms and achieving a comparable performance with the centralized benchmark have been empirically established.

BEACON has demonstrated the intimate connection between MP-MAB and CMAB, two largely disjoint sub-fields of the MAB research. It is our hope that this work sparks future interest in investigating this fundamental connection and improving existing algorithms in both areas.

## Acknowledgement

The CSs acknowledge the funding support by the US National Science Foundation under Grants ECCS-2029978, ECCS-2033671, and CNS-2002902, and the Bloomberg Data Science Ph.D. Fellowship. WX acknowledges the funding support by the Hong Kong Ph.D. Fellowship. The work of JY

was supported by the US National Science Foundation under Grants CNS-1956276, CNS-2003131, CNS-2114542, and ECCS-2030026.

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
