# Supplementary Material for "Heterogeneous Multi-player Multi-armed Bandits: Closing the Gap and Generalization"

## Chengshuai Shi, Wei Xiong, Cong Shen, and Jing Yang

## A    MP-MAB and CMAB

The formulation of the MP-MAB model in the main paper shares several similarities with the CMAB model (Chen et al., 2013, 2016b; Kveton et al., 2015c). However, these connections are largely ignored and unexplored in the previous literature, and we elaborate their similarities and differences here. First, the $K$ arms with different utilities for $M$ players can be equivalently interpreted as $MK$ base arms in the CMAB model. The matching set $\mathcal{S}$ can be viewed as one special set of super arms in CMAB, where each super arm is of size $M$ and must contain one arm from each player's $K$ arms. Furthermore, the semi-bandit feedback in CMAB assumes that observations from pulled arms are observable instead of the entire reward function, which is similar to the collision-sensing feedback discussed in the main paper. At last, the definition of reward function and regret also fit in the CMAB framework.

The key differences between MP-MAB and CMAB are in the structure of decentralized players. In CMAB, there is one centralized agent who decides all the actions and gets all the observations. However, MP-MAB is a decentralized setup where each player makes her own decisions and gets her own observations. From the perspective of decision making, the centralized configuration is more efficient as it will naturally choose the collision-free matchings. On the other hand, collision-avoidance is much harder in MP-MAB due to the decentralized decision making. To be more specific about the difference regarding the feedback, at time $t$, the centralized agent in CMAB makes decision based on the entire history $H(t) = \left\{ s_m(\tau), O_{s_m(\tau),m}(\tau) \right\}_{m \in [M], 1 \leq \tau \leq t-1}$, while player $m$ in MP-MAB makes decision with her individual history $H_m(t) = \left\{ s_m(\tau), O_{s_m(\tau),m}(\tau), \eta_{s_m(\tau)}(S(\tau)) \right\}_{1 \leq \tau \leq t-1}$. Obviously, information contained in $H_m(t)$ is more limited than that in $H(t)$. Note that $\left\{ \eta_{s_m(\tau)}(S(\tau)) \right\}_{1 \leq \tau \leq t-1}$ is omitted in $H(t)$ since it can be directly inferred by the centralized agent. Thus, MP-MAB can be viewed as a decentralized version of CMAB to some extent.

## B    Algorithmic Details of BEACON

Some omitted algorithmic details of BEACON are presented in this section.

### B.1    Orthogonalization Procedure

In the orthogonalization (sometimes also referred to as the initialization) procedure, players estimate the number of players in the MP-MAB game and obtain distinct indices in a fully distributed manner. The initialization technique from Wang et al. (2020) is adopted in BEACON. It consists of two sub-phases: orthogonalization and rank assignment. The orthogonalization sub-phase aims at assigning each player with a unique external rank $k \in [K]$. It contains a sequence of blocks with length $K + 1$, where each player attempts to fixate on arms without collision at first time step and states of fixation (successful or not) are broadcast (enabled by implicit communication). Note that in the original scheme (Wang et al., 2020), the broadcast is performed on the reserved arm $K$, which results in the need of $K > M$. To accommodate the scenarios with $K = M$, the broadcast can take place sequentially on arm 1 to arm $K$. In the rank assignment sub-phase, a modified Round-Robin sequential hopping scheme helps the players convert their external ranks to internal ranks $m \in [M]$ and estimate the overall number of players $M$. Detailed algorithms can be found in Wang et al. (2020). Using the same proofs in Lemma 1 and Lemma 2 in Wang et al. (2020), we have the following performance characterization.

**Lemma 1.** *The expected duration of the orthogonalization procedure in BEACON is less than $\frac{K^2 M}{K - M} + 2K$ time steps. Once the procedure completes, all players correctly learn the number of players $M$ and each of them is assigned with a unique index between $1$ and $M$.*

## B.2 Detailed Communication Protocols

In this section, more details of the communication design are presented. First, as illustrated in Section 3.3, the implicit communications are performed by having the "receive" player sample one arm and the "send" player either pull (create collision; bit 1) or not pull (create no collision; bit 0) the same arm to transmit one-bit information. Other players that are not communicating would fixate on other arms to avoid interruptions. The arm(s) that the players pull for receiving or avoiding are referred to as "communication arm(s)", which is an arm-player matching and is assigned before the communication happens. In BEACON, the matching of communication arms for epoch $r > 1$ is chosen as the exploration matching in the previous epoch, i.e., $S_{r-1}$. The benefit of this choice is that with the increasing explorations, $S_{r-1}$ would gradually become near-optimal with a high probability, which also leads to smaller communication losses. Specifically, in epoch $r$, follower $m > 1$ (resp. the leader) communicates to the leader (resp. follower $m > 1$) by either pulling or not pulling arm $s_1^{r-1}$ (resp. arm $s_m^{r-1}$), while the leader (resp. the follower $m$) stays on arm $s_1^{r-1}$ (resp. arm $s_m^{r-1}$) during receiving. To make this happen, in addition to the knowledge of index $s_m^{r-1}$ which is assigned to follower $m$ for explorations, index $s_1^{r-1}$ should also be communicated to the followers in the communication phase of epoch $r - 1$.

Then, as illustrated in Section 3.3, there are three kinds of information to be communicated, which are separately discussed in the following.

**Arm statistics.** The main idea of the adaptive differential communication (ADC) design is illustrated in Section 3.3. However, two important ingredients are missing. The first is when follower $m$ quantizes the arm statistics $\tilde{\mu}_{k,m}^r$ from the collected sample mean $\hat{\mu}_{k,m}^r$ using $\lceil 1 + p_{k,m}^r/2 \rceil$ bits. The least significant bit (LSB) is always ceiled to 1 if $\lceil 1 + p_{k,m}^r/2 \rceil$ bits cannot fully represent $\hat{\mu}_{k,m}^r$. We refer such process of quantizing $\tilde{\mu}_{k,m}^r$ as $\mathtt{ceil}(\hat{\mu}_{k,m}^r)$ with $\lceil 1 + p_{k,m}^r/2 \rceil$ bits. This process is needed for the later theoretical analysis to have $\tilde{\mu}_{k,m}^r \geq \hat{\mu}_{k,m}^r$.

The second missing component in ADC is referred to as the **signal-then-communicate** approach. The purpose of this approach is to synchronize the communication order and communication duration among players. It consists of two parts: the leader would first create a collision on the follower's communication arm to indicate the beginning of her statistics sharing; then, since the length of non-zero LSB at the end of $\delta_{k,m}^r$ is not fixed, after receiving the start signal, the follower $m$ would take the following approach to transmit $L$ bits ($L$ is however unknown to the leader), in which creating no collision indicates there are more bits to transmit while creating collision means the end of transmission:

$$\text{collision: start signal} \rightarrow \text{no collision} \rightarrow \text{one information bit} \rightarrow \cdots$$
$$\rightarrow \text{no collision} \rightarrow \text{one information bit} \rightarrow \text{collision: end signal.}$$

Using no collision as an indicator also reduces the practical communication loss, as it avoids creating collisions during communications. In summary, with this signal-to-communicate approach, the original $L$-bits information of arm statistics would require no more than $(2L + 2)$-bits.

**The chosen matching and leader's communication arm.** In epoch $r$, the leader needs to notify follower $m$ of both $s_m^r$ (for exploration) and $s_1^r$ (for communication in the next epoch). Similar to sharing arm statistics, the leader has to initiate the communication with a specific follower by creating a collision. Since both arm indices can be communicated via a fixed length of $\lceil \log_2(K) \rceil$ bits, they can be directly transmitted without using no-collisions to synchronize. Thus, with $K$ arms for each player, this part of communication can be done in $2\lceil \log_2(K) \rceil + 1$ bits for each follower.

**Batch size.** A naive idea to transmit the batch size $p_r$ is to directly notify the followers of this number. However, the value of $p_r$ is at most $O(\log(T))$, which requires $O(\log \log(T))$ bits. With at most $O(\log(T))$ epochs of communication, directly sharing $p_r$ may lead to a dominating regret. Luckily, sharing $p_r$ only serves to let players explore the same length, which can be achieved by a much simpler and more efficient **stop-upon-signal** approach. Specifically, while $p_r$ is calculated by the leader, rather than broadcasting it to the followers via implicit collisions, she counts the exploration length herself and creates a collision on the exploration arm of each follower upon the end of exploration in this epoch. Upon perceiving collisions, followers become aware that the current exploration phase has ended.

### B.3 Algorithm for Followers

The detailed algorithm for the follower $m$ is presented in Algorithm. 2.

---

**Algorithm 2** BEACON: Follower $m$

---

1: Set epoch counter $r \leftarrow 0$; arm counter $[p_{k,m}^r]_{k \in [K]} \leftarrow 0$; sample time $[T_{k,m}^r]_{k \in [M]} \leftarrow 0$; communicated statistics $[\tilde{\mu}_{k,m}^r]_{k \in [M]} \leftarrow 0$
2: In order $k \in [K]$, play arm $[(m-1+k) \bmod K]$ once and update sample time $T_{k,m}^{r+1} \leftarrow T_{k,m}^r + 1$
3: **while** not reaching the time horizon $T$ **do**
4:     $r \leftarrow r + 1$
5:     $\forall k \in [K], p_{k,m}^r \leftarrow \lfloor \log_2(T_{k,m}^r) \rfloor$
6:     Update $\hat{\mu}_{k,m}^r$ as the sample mean from the first $2^{p_{k,m}^r}$ exploratory samples from arm $k$
    ▷ *Communication Phase*
7:     **for** $k \in [K]$ **do**
8:         **if** $p_{k,m}^r > p_{k,m}^{r-1}$ **then**
9:             $\tilde{\mu}_{k,m}^r \leftarrow \texttt{ceil}(\hat{\mu}_{k,m}^r)$ with $\lceil 1 + p_{k,m}^r/2 \rceil$ bits
10:             $\tilde{\delta}_{k,m}^r \leftarrow \tilde{\mu}_{k,m}^r - \tilde{\mu}_{k,m}^{r-1}$
11:             $\texttt{Send}(\tilde{\delta}_{k,m}^r, 1)$
12:         **else**
13:             $\tilde{\mu}_{k,m}^r \leftarrow \tilde{\mu}_{k,m}^{r-1}$
14:         **end if**
15:     **end for**
16:     $s_m^r \leftarrow \texttt{Receive}(s_m^r, 1)$
    ▷ *Exploration Phase*
17:     Play arm $s_m^r$ until signaled
18:     Update $T_{s_m^r,m}^{r+1} \leftarrow T_{s_m^r,m}^r + 2^{p_r}$
19: **end while**

---

### B.4 Sending and Receiving Protocols

The `Send()` and `Receive()` functions in Algorithms 1 and 2 denote the protocols of sending and receiving information via forced collisions. In order to make this work self-contain, these two functions are illustrated in Algorithms 3 and 4, while a more detailed illustration of the implicit communication approach can be found in Boursier and Perchet (2019). We further note that to better expose the sending and receiving structure, Algorithms 3 and 4 contain the key ideas in implicit communications, but omit some detailed protocols, e.g., the signal-then-communicate approach.

---

**Algorithm 3** `Send()` for Player $m$

---

**Input:** bit string $\boldsymbol{u} = [u_1, u_2, ..., u_l]$ with length $l$, receiver index $n$
1: Initialization: player $m$'s communication arm $c_m$, player $n$'s communication arm $c_n$
2: **for** $i = 1, 2, \cdots, l$ **do**
3:     **if** $u_i = 1$ **then**
4:         Pull arm $c_n$     ▷ collision for bit 1
5:     **else**
6:         Pull arm $c_m$     ▷ no collision for bit 0
7:     **end if**
8: **end for**

---

**Algorithm 4** `Receive()` for Player $n$

---

**Input:** bit string $\boldsymbol{u}'$ with length $l$, sender index $m$
1: Initialization: player $n$'s communication arm $c_n$
2: **for** $i = 1, 2, \cdots, l$ **do**
3:     Pull arm $c_n$
4:     **if** collision **then**
5:         $u_i' \leftarrow 1$     ▷ collision for bit 1
6:     **else**
7:         $u_i' \leftarrow 0$     ▷ no collision for bit 0
8:     **end if**
9: **end for**
**Output:** $\boldsymbol{u}'$

---

## C Reward Functions

### C.1 Additional Examples

Other than the proportional fairness function and minimal reward function gliven in the main paper, the following general (nonlinear) reward functions are also commonly adopted in real-world applications:

- **Threshold**: $V(S,t) = \sum_{m \in [M]} \mathbb{1}\left\{O_{s_m,m}(t) \geq \varphi_m\right\}$, where $\varphi_m$ is a player-dependent threshold. It characterizes the need of reaching certain thresholds, e.g., quality-of-service requirements, in cognitive radio systems;
- **Video quality-rate model**: $V(S,t) = \sum_{m \in [M]} U_m(O_{s_m,m}(t))$, where $U_m(O_{s_m,m}(t))$ is a piecewise linear concave function on $[0,1]$ with decreasing slopes. It is typically used to describe video quality, and illustrates the decreasing of marginal utility with increased allocated resources;
- **Top-$L$ utility**: $V(S,t) = \max\left\{\sum_{m \in \mathcal{L}} O_{s_m(t),m}(t) | \mathcal{L} = [m_1,...,m_L] \subseteq [M], |\mathcal{L}| = L\right\}$, which features the highest sum of observations from any $L$ players.

### C.2  Comparison with Max-Min Fairness in Bistritz et al. (2020)

In Bistritz et al. (2020), fairness is considered among the players in MP-MAB with a specific "Max-Min" fairness measure, which shares some similarities with the minimal reward function considered in this work but with major differences discussed in the following.

**Reward function of Bistritz et al. (2020).** The instantaneous system reward gained by the players of playing matching $S$ at time $t$ in Bistritz et al. (2020) is defined as

$$V'(S,t) = \min_{m \in [M]} \left\{\mathbb{E}\left[O_{s_m,m}(t)\right]\right\} = \min\left\{\boldsymbol{\mu}_S \odot \boldsymbol{\eta}_S\right\},$$

where expectations have already been taken *inside* the minimal function. To be consistent with the notation of this paper, the corresponding expected system reward of Bistritz et al. (2020) can be written as

$$V'_{\boldsymbol{\mu},S} = \mathbb{E}\left[V'(S,t)\right] = \min_{m \in [M]} \left\{\boldsymbol{\mu}_S \odot \boldsymbol{\eta}_S\right\} = \min\left\{\boldsymbol{\mu}_S \odot \boldsymbol{\eta}_S\right\} = V'(S,t), \tag{5}$$

which does not differ from the instantaneous reward and remains the same with different utility distributions.

**Reward function of this paper.** However, for the minimal reward function defined in this work, the instantaneous reward is

$$V(S,t) = \min_{m \in [M]} \left\{O_{s_m,m}(t)\right\},$$

which is determined entirely by the instantaneously realized observations of players and does not incorporate any form of expectation. Further, the expected system reward is

$$V_{\boldsymbol{\mu},S} = \mathbb{E}\left[V(S,t)\right] = \mathbb{E}\left[\min_{m \in [M]} \left\{O_{s_m,m}(t)\right\}\right],$$

which does not have a uniform expression for different utility distributions.

**Illustration of the differences.** The differences can be illustrated more clearly by assuming that the utility distributions are mutually independent Bernoulli distributions, i.e., $\phi_{k,m} = \text{Bernoulli}(\mu_{k,m})$, where $\mu_{k,m} \leq 1$ here is the probability that utility 1 is generated by arm $(k,m)$. Then, the expected system reward function of Bistritz et al. (2020) and this work are shown in the following, respectively:

$$\text{Max-Min fairness in Bistritz et al. (2020): } V'_{\boldsymbol{\mu},S} = \min_{m \in [M]} \left\{\mu_{s_m,m}\right\};$$

$$\text{Minimal reward function in this work: } V_{\boldsymbol{\mu},S} = \prod_{m \in [M]} \mu_{s_m,m}.$$

Although the Max-Min fairness measure has several distinctions with the minimal reward function, its expected system reward function in Eqn. (5) also satisfies Assumptions 1–3. Thus, if we directly take Eqn. (5) as the expected sysmtem reward function (without explicitly defining the instantaneous reward function), both the design and analysis of BEACON are applicable to the Max-Min fairness setting in Bistritz et al. (2020). In this sense, Bistritz et al. (2020) studied a special case of the general framework proposed in this work. Furthermore, since Theorem 3 holds for this special case, this work improves the $O(\log\log(T)\log(T))$ regret provided by Bistritz et al. (2020) into a strictly $O(\log(T))$ regret.

# D Experiment Details and Additional Results

## D.1 Codes and Computational Resources

The codes for the experiments are publicly available at `https://github.com/ShenGroup/MPMAB_BEACON`, along with detailed instructions. The experiments do not require heavy computations and all the simulations were performed by a common PC, which only took a few hours to complete in total.

## D.2 Detailed Experiment Settings

All experimental results are averaged over 100 independent runs and the utility distributions are taken as mutually independent Bernoulli distributions, i.e., $\phi_{k,m} = \text{Bernoulli}(\mu_{k,m})$.. The 5-arms-5-players game adopted for the evaluation of the linear reward function shown in Fig. 2(a) is specified in the following, which is the same as the one adopted in Boursier et al. (2020):

$$\boldsymbol{\mu}^T = [\mu_{k,m}]^T_{(k,m)} = \begin{bmatrix} 0.5 & 0.49 & 0.39 & 0.29 & 0.5 \\ 0.5 & 0.49 & 0.39 & 0.29 & 0.19 \\ 0.29 & 0.19 & 0.5 & 0.499 & 0.39 \\ 0.29 & 0.49 & 0.5 & 0.5 & 0.39 \\ 0.49 & 0.49 & 0.49 & 0.49 & 0.5 \end{bmatrix}.$$

The 8-arms-6-players instance used in the simulation with the proportional fairness function and the minimal function in Figs. 2(c) and 2(d) is shown in the following:

$$\boldsymbol{\mu}^T = [\mu_{k,m}]^T_{(k,m)} = \begin{bmatrix} 0.45 & 0.49 & 0.59 & 0.17 & 0.37 & 0.86 & 0.94 & 0.98 \\ 0.39 & 0.25 & 0.4 & 0.6 & 0.24 & 0.54 & 0.43 & 0.67 \\ 0.39 & 0.33 & 0.8 & 0.01 & 0.12 & 0.2 & 0.61 & 0.77 \\ 0.95 & 0.22 & 0.24 & 0.88 & 0.2 & 0.12 & 0.29 & 0.3 \\ 0.69 & 0.89 & 0.25 & 0.59 & 0.43 & 0.18 & 0.01 & 0.84 \\ 0.97 & 0.15 & 0.89 & 0.16 & 0.09 & 0.57 & 0.61 & 0.19 \end{bmatrix}.$$

## D.3 METC Enhancements

To have a more fair comparison with METC (Boursier et al., 2020), several enhancements and adjustments are conducted. First, all empirical enhancements introduced in the supplementary material of Boursier et al. (2020) are implemented to achieve the best performance. Second, since METC is originally designed only for the linear reward function, enhancements are made to accommodate the adoption of general nonlinear reward functions. Specifically, for each active arm $(k,m)$, METC selects the empirically best matching $B_{k,m}$ containing arm $(k,m)$ w.r.t. the upper confidence bounds $\bar{\boldsymbol{\mu}}' = [\bar{\mu}'_{k,m}]_{(k,m)\in[K]\times[M]}$. The construction of $\bar{\boldsymbol{\mu}}'$ strictly follows the design from Boursier et al. (2020). In its original form, this step is confined to the linear reward function as

$$B_{k,m} \leftarrow \arg\max_{S\in\mathcal{S}, s_m=k} \left\{ \sum_{n\in[M]} \bar{\mu}'_{s_n,n} \right\}.$$

We apply the same principle to the general reward functions by assuming an enhanced oracle such that

$$B_{k,m} \leftarrow \texttt{OracleEnhanced}(\bar{\boldsymbol{\mu}}', k, m) \leftarrow \arg\max_{S\in\mathcal{S}, s_m=k} \left\{ v(\bar{\boldsymbol{\mu}}'_S \odot \boldsymbol{\eta}_S) \right\}.$$

The same idea is applied to the procedure of eliminating arms in METC. Note that the requirement for this oracle is much higher than the one used in BEACON, since it needs to output a specific exploration matching for each active arm, instead of only one matching as in BEACON.

## D.4 Additional Experimental Results

First, Figs. 3(a) and 3(b) are the complete versions of Figs. 2(a) and 2(d), where the significant advantage of BEACON over METC is illustrated more clearly.

Then, Fig. 4(a) presents the regret differences between BEACON and METC corresponding to Fig. 2(b). A large game setting with $M = 10, K = 30, T = 10^6$ are evaluated using 100 randomly generated instances with results reported in Fig. 4(b) and 4(c). We can observe that the performance

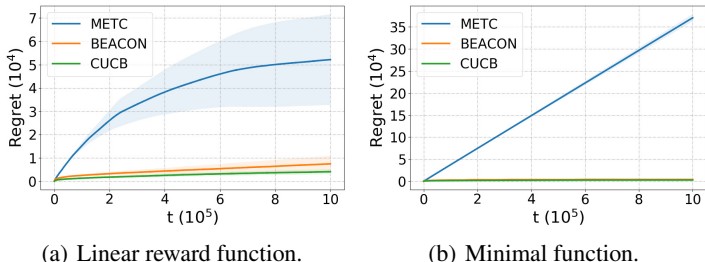

(a) Linear reward function.  (b) Minimal function.

Figure 3: Complete regret comparisons of Figs. 2(a) and 2(d). The regret curves of CUCB and BEACON are sometimes too close to each other to be distinguished.

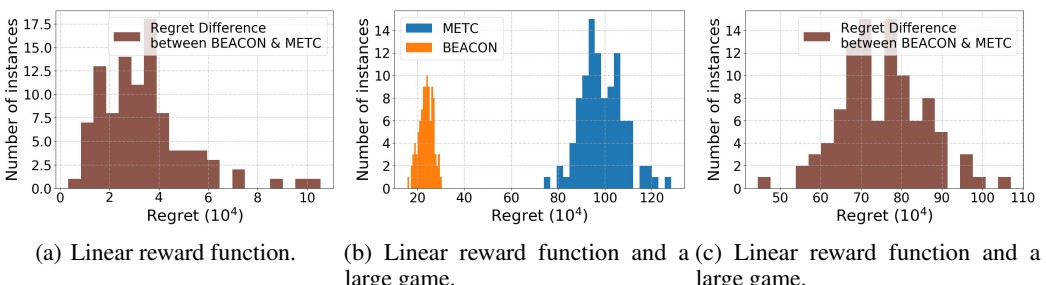

(a) Linear reward function.  (b) Linear reward function and a (c) Linear reward function and a
large game.                 large game.

Figure 4: Regret histograms with the linear reward function. (a) is the regret difference corresponding to Fig. 2(b). (b) is the cumulative result from 100 randomly generated instances with a large game setting with $M = 10$ and $K = 30$, and (c) is the corresponding regret difference for (b).

of BEACON is stable with this large game setting and is still significantly better than METC, which further demonstrates the advantages of BEACON.

Also, 100 randomly generated instances with $M = 5, K = 6, T = 10^6$ are used to evaluate the performance of BEACON and METC in dealing with the proportional fairness function. The histogram of the regrets is given in Fig. 5(a) along with the histogram of the regret differences in Fig. 5(b), the latter of which gives a more definitive illustration of the advantage of BEACON. It can be observed that BEACON effectively deals with this proportional fairness function and outperforms METC uniformly across all realizations, which again proves the stable performance of BEACON in dealing with general reward functions.

In addition to the theoretical comparison of regret analyses given in Table 1, we also provide some empirical explanations of BEACON's advantages over METC. First, the differential communication design is the key to lower communication losses. In fact, in the experiments, the statistical difference to be communicated, i.e., $\delta_{k,m}^T$, is much smaller than the theory dictates. We have frequently observed

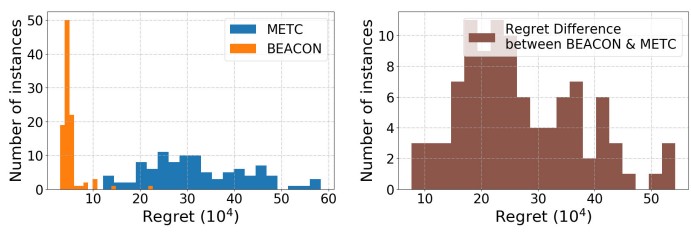

(a) Proportional fairness function.  (b) Proportional fairness function.

Figure 5: Regret histograms with the proportional fairness function. (a) is the cumulative regret result from 100 randomly generated instances, and (b) is the corresponding regret difference for (a).

that there are only one to two non-zero bits to be communicated. Second, for explorations, METC adopts the strategy of arm elimination, while BEACON does not explicitly eliminate arms but instead uses confidence bounds to balance exploration and exploitation. From the experimental results, the arm elimination approach in METC is more costly than the exploration strategy in BEACON. This improvement again illustrates the importance of the connection between MP-MAB and CMAB.

## E    Proof for Theorem 3

We begin with the analysis of BEACON with general reward functions, i.e., Theorem 3, since it is more intuitive than the one for the linear reward function, i.e., Theorem 1. The latter follows the same spirit of the former but is carefully tailored to the linear reward function.

The complete version of Theorem 3 is first presented in the following.

**Theorem 4** (**Complete version of Theorem 3**). *Under Assumptions 1, 2, and 3, the regret of BEACON is upper bounded as*

$$R(T) \leq \sum_{(k,m) \in [K] \times [M]} \left[ \frac{28 \Delta_{\min}^{k,m} \ln(T)}{(f^{-1}(\Delta_{\min}^{k,m}))^2} + \int_{\Delta_{\min}^{k,m}}^{\Delta_{\max}^{k,m}} \frac{28 \ln(T)}{(f^{-1}(x))^2} \mathrm{d}x + 4KM\Delta_{\max}^{k,m} \right]$$

$$+ \frac{6}{\ln 2} M^2 K \log_2(K) \Delta_c \ln(T) + \frac{18}{\ln 2} MK\Delta_c \ln(T) + MK\Delta_c + \left( \frac{K^2 M}{K - M} + 2K \right) \Delta_c + K\Delta_{\max}$$

$$= \tilde{O} \left( \sum_{(k,m) \in [K] \times [M]} \left[ \frac{\Delta_{\min}^{k,m}}{(f^{-1}(\Delta_{\min}^{k,m}))^2} + \int_{\Delta_{\min}^{k,m}}^{\Delta_{\max}^{k,m}} \frac{1}{(f^{-1}(x))^2} \mathrm{d}x \right] \log(T) + M^2 K\Delta_c \log(T) \right)$$

$$= \tilde{O} \left( \sum_{(k,m) \in [K] \times [M]} \frac{\Delta_{\max}^{k,m} \log(T)}{(f^{-1}(\Delta_{\min}^{k,m}))^2} + M^2 K\Delta_c \log(T) \right).$$

To facilitate the proof, we introduce (or recall) the following notations:

$V_{\boldsymbol{\mu},*} = \max\{V_{\boldsymbol{\mu},S} | S \in \mathcal{S}\} = \max\{v(\boldsymbol{\mu}_S \odot \boldsymbol{\eta}_S) | S \in \mathcal{S}\}$: the optimal reward value;

$\mathcal{S}_* = \{S | S \in \mathcal{S}, V_{\boldsymbol{\mu},S} = V_{\boldsymbol{\mu},*}\}$: the set of the optimal matchings;

$\mathcal{S}_c = \{S | \exists m \neq n, s_m = s_n\}$: the set of matchings with collisions;

$\mathcal{S}_b = \mathcal{S} \backslash (\mathcal{S}_* \cup \mathcal{S}_c)$: the set of collision-free suboptimal matchings;

$\Delta_{\min}^{k,m} = V_{\boldsymbol{\mu},*} - \max\{V_{\boldsymbol{\mu},S} | S \in \mathcal{S}_b, s_m = k\}$;

$\Delta_{\max}^{k,m} = V_{\boldsymbol{\mu},*} - \min\{V_{\boldsymbol{\mu},S} | S \in \mathcal{S}_b, s_m = k\}$;

$\Delta_{\min} = \min\{\Delta_{\min}^{k,m}\}$: the smallest reward gap among collision-free matchings;

$\Delta_{\max} = \max\{\Delta_{\max}^{k,m}\}$: the largest reward gap among collision-free matchings;

$\Delta_c = V_{\boldsymbol{\mu},*} - \min\{V_{\boldsymbol{\mu},S} | S \in \mathcal{S}_c\} \leq f(1)$: the largest possible per-step loss upon collisions.

*Proof for Theorems 3 and 4.* The overall regret $R(T)$ can be decomposed into three parts: the exploration regret $R_e(T)$, the communication regret $R_c(T)$, and the other regret $R_o(T)$, i.e.,

$$R(T) = R_e(T) + R_c(T) + R_o(T).$$

The exploration regret $R_e(T)$ and the communication regret $R_c(T)$ are caused by exploration and communication phases, respectively, and are analyzed in the following subsections. The other regret $R_o(T)$ contains the regret caused by orthogonalization and activation, i.e., the explorations before epoch 1, and can be easily bounded as

$$R_o(T) \leq \left( \frac{K^2 M}{K - M} + 2K \right) \Delta_c + K\Delta_{\max}, \tag{6}$$

where the first term is the regret from orthogonalization (Lemma 1) and the second term is the regret from activation.

With Lemmas 2 and 3, which bound $R_c(T)$ and $R_e(T)$ respectively, established in the following subsections, and the bound on $R_o(T)$ in Eqn. (6), Theorems 3 and 4 can be directly proved.    □

### E.1 Communication Regret

**Lemma 2.** *For BEACON, under time horizon $T$, the cumulative length of all communication phases $D_c$ is bounded as*

$$\mathbb{E}[D_c] \leq \frac{6}{\ln 2} M^2 K \log_2(K) \ln(T) + \frac{18}{\ln 2} MK \ln(T) + MK,$$

*and the communication loss $R_c(T)$ is bounded as*

$$R_c(T) \leq \mathbb{E}[D_c]\Delta_c \leq \frac{6}{\ln 2} M^2 K \log_2(K)\Delta_c \ln(T) + \frac{18}{\ln 2} MK\Delta_c \ln(T) + MK\Delta_c.$$

*Proof for Lemma 2.* As illustrated in Section 3.3 and Appendix B, communication phases consist of three parts of information sharing: arm statistics $\tilde{\mu}^r_{k,m}$, the chosen matching $S_r$, and the batch size parameter $p_r$. With the detailed communication protocol described in Appendix B, we bound the communication lengths of the aforementioned three parts, respectively.

**Part I: Arm statistics.** We take arm $(k,m), m \neq 1$ as an example. In epoch 1, $\tilde{\mu}^0_{k,m}$ is initialized as $0$ while $\bar{\mu}^1_{k,m}$ is the value of one random utility sample from arm $(k,m)$. With $p^1_{k,m} = \lfloor \log_2(T^1_{k,m}) \rfloor = \lfloor \log_2(1) \rfloor = 0$, $\tilde{\mu}^1_{k,m}$ is quantized from $\hat{\mu}^1_{k,m}$ with $1 + p^1_{k,m} = 1$ bit. The difference $\tilde{\delta}^1_{k,m} = \tilde{\mu}^1_{k,m} - \tilde{\mu}^0_{k,m} = \tilde{\mu}^1_{k,m}$ is transmitted and it contains only 1 bit.

In epoch $r > 1$, if $p^r_{k,m} > p^{r-1}_{k,m}$, i.e., $p^r_{k,m} = p^{r-1}_{k,m} + 1$, arm statistics of arm $(k,m)$ should be communicated via the truncated version of the difference $\tilde{\delta}^r_{k,m} = \tilde{\mu}^r_{k,m} - \tilde{\mu}^{r-1}_{k,m}$. Then, we can bound the duration of communication through bounding $\tilde{\delta}^r_{k,m}$. Specifically, it holds that

$$
\begin{aligned}
|\tilde{\delta}^r_{k,m}| &= |\tilde{\mu}^r_{k,m} - \tilde{\mu}^{r-1}_{k,m}| \\
&= |\tilde{\mu}^r_{k,m} - \hat{\mu}^r_{k,m} - (\tilde{\mu}^{r-1}_{k,m} - \hat{\mu}^{r-1}_{k,m}) + (\hat{\mu}^r_{k,m} - \hat{\mu}^{r-1}_{k,m})| \\
&\leq |\tilde{\mu}^r_{k,m} - \hat{\mu}^r_{k,m}| + |\tilde{\mu}^{r-1}_{k,m} - \hat{\mu}^{r-1}_{k,m}| + |\hat{\mu}^r_{k,m} - \hat{\mu}^{r-1}_{k,m}| \\
&\overset{(a)}{\leq} \sqrt{\frac{1}{2^{p^r_{k,m}}}} + \sqrt{\frac{1}{2^{p^r_{k,m}-1}}} + |\hat{\mu}^r_{k,m} - \hat{\mu}^{r-1}_{k,m}|,
\end{aligned}
$$

where inequality (a) is due to the quantization process specified Section B.2, i.e., $\tilde{\mu}^r_{k,m} = \texttt{ceil}(\hat{\mu}^r_{k,m})$ with $\lceil 1 + p^r_{k,m}/2 \rceil$ bits. This quantization leads to a quantization error of at most $2^{-p^r_{k,m}/2}$. Further, denoting $\gamma^{k,m}_\tau$ as the $\tau$-th random utility sample from arm $(k,m)$ during exploration phases, we can rewrite the difference $\hat{\mu}^r_{k,m} - \hat{\mu}^{r-1}_{k,m}$ as

$$
\begin{aligned}
\hat{\mu}^r_{k,m} - \hat{\mu}^{r-1}_{k,m} &= \frac{\sum_{\tau=1}^{2^{p^r_{k,m}}} \gamma^{k,m}_\tau}{2^{p^r_{k,m}}} - \frac{\sum_{\tau=1}^{2^{p^r_{k,m}-1}} \gamma^{k,m}_\tau}{2^{p^r_{k,m}-1}} \\
&= \frac{\sum_{\tau=1}^{2^{p^r_{k,m}-1}} \gamma^{k,m}_\tau + \sum_{\tau=1+2^{p^r_{k,m}-1}}^{2^{p^r_{k,m}}} \gamma^{k,m}_\tau}{2^{p^r_{k,m}}} - \frac{\sum_{\tau=1}^{2^{p^r_{k,m}-1}} \gamma^{k,m}_\tau}{2^{p^r_{k,m}-1}} \\
&= \frac{\sum_{\tau=1+2^{p^r_{k,m}-1}}^{2^{p^r_{k,m}}} \gamma^{k,m}_\tau - \sum_{\tau=1}^{2^{p^r_{k,m}-1}} \gamma^{k,m}_\tau}{2^{p^r_{k,m}}} \\
&= \frac{1}{2^{p^r_{k,m}}} \sum_{\tau=1}^{2^{p^r_{k,m}-1}} \left( \gamma^{k,m}_{\tau+2^{p^r_{k,m}-1}} - \gamma^{k,m}_\tau \right)
\end{aligned}
$$

which is a $\frac{1}{\sqrt{2^{p^r_{k,m}+1}}}$-sub-Gaussian random variable since the utility samples are independent across time. Thus, we can further derive that, with a dummy variable $x \geq \sqrt{\ln 2}$,

$$\mathbb{P}\left( \left| \hat{\mu}^r_{k,m} - \hat{\mu}^{r-1}_{k,m} \right| \geq \sqrt{\frac{x^2}{2^{p^r_{k,m}}}} \right) \leq 2\exp\left[ -2^{p^r_{k,m}} \frac{x^2}{2^{p^r_{k,m}}} \right] \leq 2\exp[-x^2]$$

$$\Rightarrow \mathbb{P}\left(|\tilde{\delta}_{k,m}^r| \geq \sqrt{\frac{1}{2^{p_{k,m}^r}}} + \sqrt{\frac{1}{2^{p_{k,m}^r-1}}} + \sqrt{\frac{x^2}{2^{p_{k,m}^r}}}\right) \leq 2\exp[-x^2]$$

$$\overset{(a)}{\Rightarrow} \mathbb{P}\left(L_{k,m}^r \geq 3 + \frac{p_{k,m}^r}{2} + \log_2\left(\frac{1+\sqrt{2}+x}{\sqrt{2^{p_{k,m}^r}}}\right)\right) \leq 2\exp[-x^2]$$

$$\Rightarrow \mathbb{P}\left(L_{k,m}^r \geq 3 + \log_2(3+x)\right) \leq 2\exp[-x^2]$$

$$\Rightarrow \mathbb{P}\left(L_{k,m}^r \leq 3 + \log_2(3+x)\right) \geq 1 - 2\exp[-x^2]$$

$$\overset{(b)}{\Rightarrow} \mathbb{P}\left(L_{k,m}^r \leq l\right) \geq 1 - 2\exp\left[-(2^{l-3}-3)^2\right]$$

where $L_{k,m}^r$ in implication (a) is the length of the truncated version $|\tilde{\delta}_{k,m}^r|$ and is upper bounded by

$$L_{k,m}^r \leq \lceil 1 + p_{k,m}^r/2 \rceil - \lfloor \log_2(1/|\tilde{\delta}_{k,m}^r|) \rfloor$$
$$\leq 3 + p_{k,m}^r/2 + \log_2(|\tilde{\delta}_{k,m}^r|).$$

In deriving (b), we substitute the variable $3 + \log_2(3+x)$ with $l$, which satisfies that $l \geq 3 + \log_2(3 + \sqrt{\ln 2})$, and thus equivalently $x = 2^{l-3} - 3$. With the above results and viewing $L_{k,m}^r$ as a random variable, we have that its cumulative distribution function (CDF) $F_{L_{k,m}^r}(l)$ satisfies the following property:

$$\forall l \geq 5 > 3 + \log_2(3 + \sqrt{\ln 2}), F_{L_{k,m}^r}(l) = \mathbb{P}\left(L_{k,m}^r \leq l\right) \geq 1 - 2\exp\left[-(2^{l-3}-3)^2\right].$$

Using the property of CDF, we can bound the expectation of $L_{k,m}^r$ as

$$\mathbb{E}\left[L_{k,m}^r\right] = \sum_{l=0}^{\infty}(1 - F_{L_{k,m}^r}(l))$$
$$\leq 6 + \sum_{l=6}^{\infty} 2\exp\left[-(2^{l-3}-3)^2\right]$$
$$\leq 6 + \int_{l=5}^{\infty} 2\exp\left[-(2^{l-3}-3)^2\right]\mathrm{d}l$$
$$\leq 7.$$

Thus, we have that in expectation, the truncated version of $|\tilde{\delta}_{k,m}^r|$ has a length that is less than 7 bits. In addition, 1-bit information should also be transmitted to indicate the sign of $\tilde{\delta}_{k,m}^r$. As a summary, in expectation, 8 bits is sufficient to represent the truncated version of $\tilde{\delta}_{k,m}^r$,

With overall time horizon of $T$, there are at most $\log_2(T)$ statistics updates of arm $(k,m)$ in addition to the first epoch. The expected communication duration for arm statistics $D_s$ is bounded as

$$\mathbb{E}\left[D_s\right] \overset{(a)}{=} \underbrace{MK}_{\text{epoch } r\,=\,1} + \mathbb{E}\left[\underbrace{\sum_r \sum_{(k,m):p_{k,m}^r > p_{k,m}^{r-1}} (2 + 2(L_{k,m}^r + 1))}_{\text{epoches } r\,>\,1}\right]$$
$$\leq MK + (2 + 2\times 8)MK\log_2(T)$$
$$\leq 18MK\log_2(T) + MK$$
$$= \frac{18}{\ln 2}MK\ln(T) + MK, \tag{7}$$

where equation (a) takes the signal-then-communicate protocol described in Appendix B into consideration, where transmitting $\tilde{\delta}_{k,m}^r$ consists of 1 step of the leader notifying the follower to start, $(L_{k,m}^r+1)$ steps of the truncated version of $\tilde{\delta}_{k,m}^r$ and correspondingly $(L_{k,m}^r + 2)$ steps of synchronization between the leader and follower.

**Part II & III: Matching choice and batch size.** These two parts of communications are relatively easy to bound. In each epoch $r$, the leader initiates and then transmits two arm indices ($s_1^r$ and $s_m^r$) to each follower $m$, thus, the communication duration $D_m$ for matching assignments is bounded as

$$D_m = \sum_r (M-1)(1 + 2\lceil \log_2(K) \rceil)$$

$$\leq (M-1)(2\log_2(K) + 3)MK\log_2(T)$$

$$< \frac{1}{\ln 2}M^2 K(2\log_2(K) + 3)\ln(T). \tag{8}$$

For the communication duration $D_b$ for the batch size, as illustrated in Appendix B, the leader notifies followers to stop exploring by sending stopping signals. Thus, it holds that

$$D_b = \sum_r (M-1) \leq (M-1)MK\log_2(T) < \frac{1}{\ln 2}M^2 K\ln(T). \tag{9}$$

By combining Eqns. (7), (8) and (9), Lemma 2 can be obtained as

$$\mathbb{E}[D_c] = \mathbb{E}[D_s] + \mathbb{E}[D_m] + \mathbb{E}[D_b]$$

$$\leq \frac{18}{\ln 2}MK\ln(T) + MK + \frac{1}{\ln 2}M^2(2\log_2(K) + 3)K\ln(T) + \frac{1}{\ln 2}M^2 K\ln(T)$$

$$\leq \frac{6}{\ln 2}M^2 K\log_2(K)\ln(T) + \frac{18}{\ln 2}MK\ln(T) + MK.$$

$\square$

### E.2 Exploration Regret

**Lemma 3.** *For BEACON, under time horizon $T$, the exploration regret is upper bounded as*

$$R_e(T) \leq \sum_{(k,m)\in[K]\times[M]} \left[ \frac{28\Delta_{\min}^{k,m}\ln(T)}{(f^{-1}(\Delta_{\min}^{k,m}))^2} + \int_{\Delta_{\min}^{k,m}}^{\Delta_{\max}^{k,m}} \frac{28\ln(T)}{(f^{-1}(x))^2}\mathrm{d}x + 4KM\Delta_{\max}^{k,m} \right].$$

*Proof for Lemma 3.* The following proof is inspired by the proof for CUCB in Chen et al. (2013). However, Chen et al. (2013) does not consider the batched structure, which introduces additional challenges for the proof here. To better characterize the exploration regret, we introduce the following notations:

$$\mathcal{S}_b^{k,m} = \{S|S \in \mathcal{S}_b, s_m = k\} = \{S_1^{k,m}, ..., S_{N(k,m)}^{k,m}\};$$

$$\Delta_n^{k,m} = V_{\boldsymbol{\mu},*} - V_{\boldsymbol{\mu}, S_n^{k,m}}, \forall n \in \{1, ..., N(k,m)\},$$

where $\mathcal{S}_b^{k,m}$ is the set of collision-free sub-optimal matchings that contain arm $(k,m)$ and we denote its size as $N(k,m)$. $\Delta_n^{k,m}$ denotes the sub-optimality gap of the matching $S_n^{k,m}$. In the following proof, we re-arrange the set $\mathcal{S}_b^{k,m} = \{S_1^{k,m}, ..., S_{N(k,m)}^{k,m}\}$ in a decreasing order w.r.t. the gap $\Delta_n^{k,m}$, i.e., if $n_1 \geq n_2$, $\Delta_{n_1}^{k,m} \leq \Delta_{n_2}^{k,m}$. Also, for convenience, we denote $\Delta_{N(k,m)+1}^{k,m} = 0$. Furthermore, it naturally holds that $\Delta_{\min}^{k,m} = \Delta_{N(k,m)}^{k,m}$ and $\Delta_{\max}^{k,m} = \Delta_1^{k,m}$.

We denote $q_n^{k,m}, \forall n \in \{1, ..., N(k,m)\}$ as the integer such that

$$2^{q_n^{k,m}-1} \leq \frac{14\ln(T)}{(f^{-1}(\Delta_n^{k,m}))^2} < 2^{q_n^{k,m}} < \frac{28\ln(T)}{(f^{-1}(\Delta_n^{k,m}))^2}.$$

In addition, we define $q_0^{k,m} = 0$ and $q_{N(k,m)+1}^{k,m} = \lceil \log_2(T) \rceil$. Note that with the above definition of $q_n^{k,m}$, it holds that

$$\forall p \geq q_n^{k,m}, f\left(2\sqrt{\frac{3\ln t_r}{2^{p+1}}} + \sqrt{\frac{1}{2^p}}\right) \leq f\left(3\sqrt{\frac{3\ln t_r}{2^{p+1}}}\right) \leq f\left(3\sqrt{\frac{3\ln T}{2^{p+1}}}\right) < \Delta_n^{k,m}, \tag{10}$$

which is a key property that is utilized in the subsequent proofs.

For epoch $r$, we define the "representative arm" $\rho_r = (s_m^r, m)$ as one of the arms in $S_r$ such that $p_{s_m^r, m}^r = p_r$. If there are more than one arm in $S_r$ with arm counter $p_r$, $\rho_r$ is randomly chosen from them. Thus, it is guaranteed that there is one and only one representative arm for each exploration phase. With the arm counter updating rule specified in Section 3.2, the counter of arm $\rho_r$ will certainly increase by 1 after epoch $r$.

**Step I: Regret decomposition.** With respect to the representative arm, we decompose the exploration regret as

$$
\begin{aligned}
R_e(T) &= \mathbb{E}\left[\sum_r 2^{p_r}(V_{\boldsymbol{\mu},*} - V_{\boldsymbol{\mu},S_r})\right] \\
&= \mathbb{E}\left[\sum_r \sum_{(k,m)\in[K]\times[M]} 2^{p_r}(V_{\boldsymbol{\mu},*} - V_{\boldsymbol{\mu},S_r})\mathbb{1}\left\{\rho_r = (k,m)\right\}\right] \\
&\stackrel{(a)}{=} \mathbb{E}\left[\sum_r \sum_{(k,m)\in[K]\times[M]} 2^{p_{k,m}^r}(V_{\boldsymbol{\mu},*} - V_{\boldsymbol{\mu},S_r})\mathbb{1}\left\{\rho_r = (k,m)\right\}\right] \\
&\stackrel{(b)}{=} \mathbb{E}\left[\sum_r \sum_{(k,m)\in[K]\times[M]} \sum_{n=1}^{N(k,m)} 2^{p_{k,m}^r}\Delta_n^{k,m}\mathbb{1}\left\{\rho_r = (k,m), S_r = S_n^{k,m}\right\}\right] \\
&\stackrel{(c)}{=} \mathbb{E}\left[\sum_{(k,m)\in[K]\times[M]} \sum_{p_{k,m}\geq 0} \sum_{n=1}^{N(k,m)} 2^{p_{k,m}}\Delta_n^{k,m}\mathbb{1}\left\{S_{k,m,p_{k,m}} = S_n^{k,m}\right\}\right] \\
&\stackrel{(d)}{=} \sum_{(k,m)\in[K]\times[M]} R_e^{k,m}(T),
\end{aligned}
\tag{11}
$$

where equality (a) is from the definition of the representative arm that if $\rho_r = (k,m)$, it holds that $p_r = p_{k,m}^r$. Equality (b) further associates the regret of each exploration phase with specific sub-optimal matchings. $S_{k,m,p_{k,m}}$ denotes the exploration matching with representative arm $(k,m)$ and the corresponding arm counter $p_{k,m}$. Equality (c) holds because once $\rho_r = (k,m)$, its arm counter will increase. Equality (d) denotes $R_e^{k,m}(T) :=$ $\mathbb{E}\left[\sum_{p_{k,m}>0}\sum_{n=1}^{N(k,m)} 2^{p_{k,m}}\Delta_n^{k,m}\mathbb{1}\left\{S_{k,m,p_{k,m}} = S_n^{k,m}\right\}\right]$, which represents the regret associated with arm $(k,m)$.

For term $R_e^{k,m}(T)$, we further have

$$
\begin{aligned}
R_e^{k,m}(T) &= \mathbb{E}\left[\sum_{p_{k,m}\geq 0} \sum_{n=1}^{N(k,m)} 2^{p_{k,m}}\Delta_n^{k,m}\mathbb{1}\left\{S_{k,m,p_{k,m}} = S_n^{k,m}\right\}\right] \\
&= \sum_{p_{k,m}\geq 0} \sum_{n=1}^{N(k,m)} 2^{p_{k,m}}\Delta_n^{k,m}\mathbb{P}\left(S_{k,m,p_{k,m}} = S_n^{k,m}\right) \\
&\stackrel{(a)}{\leq} \sum_{p_{k,m}>0} \sum_{n=1}^{N(k,m)} 2^{p_{k,m}}\Delta_n^{k,m}\mathbb{P}\left(S_{k,m,p_{k,m}} = S_n^{k,m}|\mathcal{E}_{k,m,p_{k,m}}\right)\mathbb{P}\left(\mathcal{E}_{k,m,p_{k,m}}\right) \\
&\quad + \sum_{p_{k,m}\geq 0} \sum_{n=1}^{N(k,m)} 2^{p_{k,m}}\Delta_n^{k,m}\mathbb{P}\left(S_{k,m,p_{k,m}} = S_n^{k,m}|\bar{\mathcal{E}}_{k,m,p_{k,m}}\right)\mathbb{P}\left(\bar{\mathcal{E}}_{k,m,p_{k,m}}\right) \\
&\leq \sum_{p_{k,m}\geq 0} \sum_{n=1}^{N(k,m)} 2^{p_{k,m}}\Delta_n^{k,m}\mathbb{P}\left(S_{k,m,p_{k,m}} = S_n^{k,m}|\mathcal{E}_{k,m,p_{k,m}}\right)
\end{aligned}
$$

$$+ \sum_{p_{k,m} \geq 0} 2^{p_{k,m}} \Delta_{\max}^{k,m} \mathbb{P}\left(\bar{\mathcal{E}}_{k,m,p_{k,m}}\right)$$

$$\leq \underbrace{\sum_{h=0}^{N(k,m)} \sum_{q_h^{k,m} \leq p_{k,m} < q_{h+1}^{k,m}} \sum_{n=1}^{N(k,m)} 2^{p_{k,m}} \Delta_n^{k,m} \mathbb{P}\left(S_{k,m,p_{k,m}} = S_n^{k,m} | \mathcal{E}_{k,m,p_{k,m}}\right)}_{\text{term (A)}}$$

$$+ \underbrace{\sum_{p_{k,m} \geq 0} 2^{p_{k,m}} \Delta_{\max}^{k,m} \mathbb{P}\left(\bar{\mathcal{E}}_{k,m,p_{k,m}}\right)}_{\text{term (B)}},$$

where equality (a) introduces the notion of the "nice event" $\mathcal{E}_{k,m,p_{k,m}}$, which is described in the following.

At epoch $r$, the nice event $\mathcal{E}_r$ is defined as

$$\mathcal{E}_r = \left\{ \forall (k,m) \in [K] \times [M], -\sqrt{\frac{3 \ln t_r}{2^{p_{k,m}^r+1}}} < \tilde{\mu}_{k,m}^r - \mu_{k,m} < \sqrt{\frac{3 \ln t_r}{2^{p_{k,m}^r+1}}} + \sqrt{\frac{1}{2^{p_{k,m}^r}}} \right\}.$$

Furthermore, when the representative arm in epoch $r$ is arm $(k,m)$ with counter $p_{k,m}$, $\mathcal{E}_r$ is denoted as $\mathcal{E}_{k,m,p_{k,m}}$.

**Step II: Bounding term (B).** We start with term (B) by bounding the probability that event $\bar{\mathcal{E}}_r$ happens. Specifically, it holds that

$$\mathbb{P}\left(\bar{\mathcal{E}}_r\right) \leq \sum_{(k,m) \in [K] \times [M]} \mathbb{P}\left(\tilde{\mu}_{k,m}^r - \mu_{k,m} \leq -\sqrt{\frac{3 \ln t_r}{2^{p_{k,m}^r+1}}}\right)$$

$$+ \sum_{(k,m) \in [K] \times [M]} \mathbb{P}\left(\tilde{\mu}_{k,m}^r - \mu_{k,m} \geq \sqrt{\frac{3 \ln t_r}{2^{p_{k,m}^r+1}}} + \sqrt{\frac{1}{2^{p_{k,m}^r}}}\right)$$

$$= \sum_{(k,m) \in [K] \times [M]} \mathbb{P}\left(\tilde{\mu}_{k,m}^r - \hat{\mu}_{k,m}^r + \hat{\mu}_{k,m}^r - \mu_{k,m} \leq -\sqrt{\frac{3 \ln t_r}{2^{p_{k,m}^r+1}}}\right)$$

$$+ \sum_{(k,m) \in [K] \times [M]} \mathbb{P}\left(\tilde{\mu}_{k,m}^r - \hat{\mu}_{k,m}^r + \hat{\mu}_{k,m}^r - \mu_{k,m} \geq \sqrt{\frac{3 \ln t_r}{2^{p_{k,m}^r+1}}} + \sqrt{\frac{1}{2^{p_{k,m}^r}}}\right)$$

$$\overset{(a)}{\leq} \sum_{(k,m) \in [K] \times [M]} \mathbb{P}\left(\hat{\mu}_{k,m}^r - \mu_{k,m} \leq -\sqrt{\frac{3 \ln t_r}{2^{p_{k,m}^r+1}}}\right)$$

$$+ \sum_{(k,m) \in [K] \times [M]} \mathbb{P}\left(\hat{\mu}_{k,m}^r - \mu_{k,m} \geq \sqrt{\frac{3 \ln t_r}{2^{p_{k,m}^r+1}}}\right)$$

$$\leq \sum_{(k,m) \in [K] \times [M]} \sum_{p_{k,m}=0}^{\lfloor \log_2(t_r) \rfloor} 2\mathbb{P}\left(\hat{\mu}_{k,m}^r - \mu_{k,m} \geq \sqrt{\frac{3 \ln t_r}{2^{p_{k,m}^r+1}}}, p_{k,m}^r = p_{k,m}\right)$$

$$\leq \sum_{(k,m) \in [K] \times [M]} \sum_{p_{k,m}=0}^{\lfloor \log_2(t_r) \rfloor} 2\mathbb{P}\left(\frac{\sum_{\tau=1}^{2^{p_{k,m}}} \gamma_\tau^{k,m}}{2^{p_{k,m}}} - \mu_{k,m} \geq \sqrt{\frac{3 \ln t_r}{2^{p_{k,m}+1}}}\right)$$

$$\overset{(b)}{\leq} \sum_{(k,m) \in [K] \times [M]} \sum_{p_{k,m}=0}^{\lfloor \log_2(t_r) \rfloor} 2 \exp\left[-2 \cdot 2^{p_{k,m}} \frac{3 \ln t_r}{2^{p_{k,m}+1}}\right]$$

$$\leq 2KM \frac{\lfloor \log_2(t_r) \rfloor + 1}{(t_r)^3}$$

$$\leq 2KM \frac{1}{(t_r)^2}$$

$$\overset{(c)}{\leq} 2KM\frac{1}{(2^{p_r})^2},\tag{12}$$

where inequality (a) holds because $\tilde{\mu}_{k,m}^r = \mathtt{ceil}(\hat{\mu}_{k,m}^r)$ with $\lceil 1+p_{k,m}^r/2\rceil$ bits and $\tilde{\mu}_{k,m}^r - \hat{\mu}_{k,m}^r > 0$. Inequality (b) is from the Hoeffding's inequality. Inequality (c) utilizes the observation that $t_r \geq 2^{p_r}$.

With Eqn. (12), we can further bound term (B) as

$$\text{term (B)} = \sum_{p_{k,m}\geq 0} 2^{p_{k,m}}\Delta_{\max}^{k,m}\mathbb{P}\left(\bar{\mathcal{E}}_{k,m,p_{k,m}}\right)$$

$$\overset{(a)}{\leq} 2\sum_{p_{k,m}\geq 0} 2^{p_{k,m}}\Delta_{\max}^{k,m}\cdot KM\frac{1}{(2^{p_{k,m}})^2}$$

$$= 2\sum_{p_{k,m}\geq 0}\Delta_{\max}^{k,m}\cdot KM\frac{1}{2^{p_{k,m}}}$$

$$\leq 4KM\Delta_{\max}^{k,m},$$

where inequality (a) is with Eqn. (12) and $p_r = p_{k,m}$.

**Step III: Bounding term (A).** Before bounding term (A), we first establish the following implications. For epoch $r$, if $\rho_r = (k,m)$ and $p_r = p_{k,m}^r = p_{k,m}$, denoting $\bar{\boldsymbol{\mu}}_r$ and $S_r$ as $\bar{\boldsymbol{\mu}}^{k,m,p_{k,m}}$ and $S_{k,m,p_{k,m}}$ respectively, if event $\mathcal{E}_{k,m,p_{k,m}}$ happens, we have

$$p_{k,m}\geq q_h^{k,m},\text{ the oracle outputs } S_{k,m,p_{k,m}} = S_n^{k,m}$$

$$\Rightarrow p_{k,m}\geq q_h^{k,m}, \forall S\in\mathcal{S}_*\backslash\mathcal{S}_c, v(\bar{\boldsymbol{\mu}}_{S_n^{k,m}}^{k,m,p_{k,m}}\odot\boldsymbol{\eta}_{S_n^{k,m}})\geq v(\bar{\boldsymbol{\mu}}_S^{k,m,p_{k,m}}\odot\boldsymbol{\eta}_S)$$

$$\Rightarrow p_{k,m}\geq q_h^{k,m}, \forall S\in\mathcal{S}_*\backslash\mathcal{S}_c, v(\bar{\boldsymbol{\mu}}_{S_n^{k,m}}^{k,m,p_{k,m}})\geq v(\bar{\boldsymbol{\mu}}_S^{k,m,p_{k,m}})$$

$$\overset{(a)}{\Rightarrow} p_{k,m}\geq q_h^{k,m}, \forall S\in\mathcal{S}_*\backslash\mathcal{S}_c, v(\boldsymbol{\mu}_{S_n^{k,m}}) + f\left(\left\|\bar{\boldsymbol{\mu}}_{S_n^{k,m}}^{k,m,p_{k,m}} - \boldsymbol{\mu}_{S_n^{k,m}}\right\|_\infty\right)\geq v(\bar{\boldsymbol{\mu}}_S^{k,m,p_{k,m}})$$

$$\overset{(b)}{\Rightarrow} p_{k,m}\geq q_h^{k,m}, \forall S\in\mathcal{S}_*\backslash\mathcal{S}_c, V_{\boldsymbol{\mu},S_n^{k,m}} + f\left(2\sqrt{\frac{3\ln t_r}{2^{p_{k,m}+1}}} + \sqrt{\frac{1}{2^{p_{k,m}}}}\right)\geq V_{\boldsymbol{\mu},*}$$

$$\overset{(c)}{\Rightarrow} p_{k,m}\geq q_h^{k,m}, V_{S_n^{k,m}} + \Delta_h^{k,m} > V_*,\tag{13}$$

where implication (a) is from Assumption 3 and implication (b) utilizes the definition of $\mathcal{E}_{k,m,p_{k,m}}$, Assumption 2 and that arms in $S_{k,m,p_{k,m}}$ have counters at least $p_{k,m}$. Implication (c) is from the definition of $q_h^{k,m}$ and Eqn. (10).

With Eqn. (13), we can get that if $p_{k,m}\geq q_h^{k,m}$, the matchings $S_n^{k,m}$ with $n\leq h$ cannot be $S_r$; otherwise it contradicts with the definition of $\Delta_h^{k,m}$. Thus, we can further bound term (A) as

$$\text{term (A)} = \sum_{h=0}^{N(k,m)}\sum_{q_h^{k,m}\leq p_{k,m}<q_{h+1}^{k,m}}\sum_{n=1}^{N(k,m)} 2^{p_{k,m}}\Delta_n^{k,m}\mathbb{P}\left(S_{k,m,p_{k,m}} = S_n^{k,m}|\mathcal{E}_{k,m,p_{k,m}}\right)$$

$$= \sum_{h=0}^{N(k,m)}\sum_{q_h^{k,m}\leq p_{k,m}<q_{h+1}^{k,m}}\sum_{n=h+1}^{N(k,m)} 2^{p_{k,m}}\Delta_n^{k,m}\mathbb{P}\left(S_{k,m,p_{k,m}} = S_n^{k,m}|\mathcal{E}_{k,m,p_{k,m}}\right)$$

$$\overset{(a)}{\leq} \sum_{h=0}^{N(k,m)}\sum_{q_h^{k,m}\leq p_{k,m}<q_{h+1}^{k,m}}\sum_{n=h+1}^{N(k,m)} 2^{p_{k,m}}\Delta_{h+1}^{k,m}\mathbb{P}\left(S_{k,m,p_{k,m}} = S_n^{k,m}|\mathcal{E}_{k,m,p_{k,m}}\right)$$

$$\overset{(b)}{\leq} \sum_{h=0}^{N(k,m)}\sum_{q_h^{k,m}\leq p_{k,m}<q_{h+1}^{k,m}} 2^{p_{k,m}}\Delta_{h+1}^{k,m}$$

$$= \sum_{h=0}^{N(k,m)} (2^{q_{h+1}^{k,m}} - 2^{q_h^{k,m}})\Delta_{h+1}^{k,m}$$

$$= \sum_{h=0}^{N(k,m)-1} (2^{q_{h+1}^{k,m}} - 2^{q_h^{k,m}}) \Delta_{h+1}^{k,m}$$

$$\leq 2^{q_{N(k,m)}^{k,m}} \Delta_{N(k,m)}^{k,m} + \sum_{h=1}^{N(k,m)-1} 2^{q_h^{k,m}} \left( \Delta_h^{k,m} - \Delta_{h+1}^{k,m} \right)$$

$$\overset{(c)}{\leq} \frac{28 \Delta_{N(k,m)}^{k,m} \ln(T)}{(f^{-1}(\Delta_{N(k,m)}^{k,m}))^2} + \sum_{h=1}^{N(k,m)-1} \frac{28 \ln(T)}{(f^{-1}(\Delta_h^{k,m}))^2} \left( \Delta_h^{k,m} - \Delta_{h+1}^{k,m} \right)$$

$$\overset{(d)}{\leq} \frac{28 \Delta_{N(k,m)}^{k,m} \ln(T)}{(f^{-1}(\Delta_{N(k,m)}^{k,m}))^2} + \int_{\Delta_{N(k,m)}^{k,m}}^{\Delta_1^{k,m}} \frac{28 \ln(T)}{(f^{-1}(x))^2} dx$$

$$= \frac{28 \Delta_{\min}^{k,m} \ln(T)}{(f^{-1}(\Delta_{\min}^{k,m}))^2} + \int_{\Delta_{\min}^{k,m}}^{\Delta_{\max}^{k,m}} \frac{28 \ln(T)}{(f^{-1}(x))^2} dx,$$

where inequality (a) holds because $\forall n \geq h+1$, $\Delta_n^{k,m} \leq \Delta_{h+1}^{k,m}$, and inequality (b) is from $\sum_{n=h+1}^{N(k,m)} \mathbb{P}\left(S_{k,m,p_{k,m}} = S_n^{k,m}|\mathcal{E}_{k,m,p_{k,m}}\right) \leq 1$. Inequality (c) is from the definition of $q_n^{k,m}$ and inequality (d) is because $\frac{28\ln(T)}{(f^{-1}(x))^2}$ is strictly decreasing in $[\Delta_{N(k,m)}^{k,m}, \Delta_1^{k,m}]$.

By combining terms (A) and (B), we have

$$R_e^{k,m}(T) \leq \frac{28\Delta_{\min}^{k,m}\ln(T)}{(f^{-1}(\Delta_{\min}^{k,m}))^2} + \int_{\Delta_{\min}^{k,m}}^{\Delta_{\max}^{k,m}} \frac{28\ln(T)}{(f^{-1}(x))^2}dx + 4KM\Delta_{\max}^{k,m}$$

$$\leq \frac{28\Delta_{\max}^{k,m}\ln(T)}{(f^{-1}(\Delta_{\min}^{k,m}))^2} + 4KM\Delta_{\max}^{k,m}.$$

Overall, we conclude that

$$R_e(T) = \sum_{(k,m)\in[K]\times[M]} R_e^{k,m}(T)$$

$$\leq \sum_{(k,m)\in[K]\times[M]} \left[ \frac{28\Delta_{\min}^{k,m}\ln(T)}{(f^{-1}(\Delta_{\min}^{k,m}))^2} + \int_{\Delta_{\min}^{k,m}}^{\Delta_{\max}^{k,m}} \frac{28\ln(T)}{(f^{-1}(x))^2}dx + 4KM\Delta_{\max}^{k,m} \right]$$

$$\leq \sum_{(k,m)\in[K]\times[M]} \frac{28\Delta_{\max}^{k,m}\ln(T)}{(f^{-1}(\Delta_{\min}^{k,m}))^2} + 4K^2M^2\Delta_{\max}.$$

$\square$

Theorems 3 and 4 can be proved by combining Lemmas 2, 3, and Eqn. (6).

## F   Proof for Theorem 1

A complete version of Theorem 1 is first presented in the following.

**Theorem 5** (**Complete version of Theorem 1**). *With a linear reward function, the regret of BEACON is upper bounded as*

$$R_{\text{linear}}(T) \leq \sum_{(k,m)\in[K]\times[M]} \frac{3727M}{\Delta_{\min}^{k,m}} \ln(T) + 8K^2M^3 + M^2K$$

$$+ (22M + 2M\log_2(K)) \left[ \frac{2MK}{\ln 2}\ln(T) + MK\left( \frac{3M\sqrt{3\ln(T)}}{\sqrt{2}-1} + \frac{8KM^2}{3} \right) \right]$$

$$= \tilde{O}\left( \sum_{(k,m)\in[K]\times[M]} \frac{M\log(T)}{\Delta_{\min}^{k,m}} + M^2K\log(T) \right)$$

$$= \tilde{O}\left(\frac{M^2 K \log(T)}{\Delta_{\min}} + M^2 K \log(T)\right).$$

*Proof for Theorems 1 and 5.* Similar to the previous proof, the overall regret $R_{\text{linear}}(T)$ can be decomposed into three parts: the exploration regret $R_{e,\text{linear}}(T)$, the communication regret $R_{c,\text{linear}}(T)$, and the other regret $R_{o,\text{linear}}(T)$, i.e.,

$$R_{\text{linear}}(T) = R_{e,\text{linear}}(T) + R_{c,\text{linear}}(T) + R_{o,\text{linear}}(T).$$

The last component can be similarly bounded as

$$R_{o,\text{linear}}(T) \leq \left(\frac{K^2 M}{K-M} + 2K\right)\Delta_c + K\Delta_{\max},$$

The communication regret and exploration regret are bounded Lemmas 4 and 5 that are presented in the subsequent subsections. Putting them all together completes the proof. $\qquad\square$

### F.1 Communication Regret

**Lemma 4.** *For BEACON, under time horizon $T$, the communication loss $R_{c,linear}(T)$ is upper bounded as*

$$R_{c,linear}(T) \leq M^2 K + (22M + 2M\log_2(K))\left[\frac{2MK}{\ln 2}\ln(T) + MK\left(\frac{3M\sqrt{3\ln(T)}}{\sqrt{2}-1} + \frac{8KM^2}{3}\right)\right].$$

*Proof for Lemma 4.* From the proof for Lemma 2, we can draw the following facts:

(i) For epoch 1, communicating $\tilde{\delta}_{k,m}^1$ takes 1 time step;

(ii) For epoch $r > 1$, if $p_{k,m}^r > p_{k,m}^{r-1}$, $\tilde{\delta}_{k,m}^r$ is communicated and the communication in expectation takes $2 + 2 \times (1 + \mathbb{E}[L_{k,m}^r]) \leq 18$ time steps;

(iii) For epoch $r > 1$, the communication of the chosen matching and the batch size parameter takes less than $M(3 + 2\log_2(K)) + M$ time steps.

These facts hold for the general reward functions, thus naturally hold for the linear reward function.

However, with the linear reward function, the loss caused by communication can be characterized more carefully as

$$R_{c,\text{linear}}(T) \stackrel{(a)}{\leq} MK \times M$$

$$+ \mathbb{E}\left[\sum_r (2 + V_{\boldsymbol{\mu},*} - V_{\boldsymbol{\mu},S_r})\mathbb{1}\{\mathcal{E}_r\}\left[\sum_{(k,m)} 18\mathbb{1}\left\{p_{k,m}^r \geq p_{k,m}^{r-1}\right\} + M(3 + 2\log_2(K)) + M\right]\right]$$

$$+ \mathbb{E}\left[\sum_r M\mathbb{1}\{\bar{\mathcal{E}}_r\}\left[\sum_{(k,m)} 18\mathbb{1}\left\{p_{k,m}^r \geq p_{k,m}^{r-1}\right\} + M(3 + 2\log_2(K)) + M\right]\right]$$

$$\stackrel{(b)}{\leq} M^2 K + \sum_r \mathbb{E}\left[(2 + V_{\boldsymbol{\mu},*} - V_{\boldsymbol{\mu},S_r})\mathbb{1}\{\mathcal{E}_r\} + M\mathbb{1}\{\bar{\mathcal{E}}_r\}\right](22M + 2M\log_2(K))$$

$$\stackrel{(c)}{\leq} M^2 K + \sum_r \left(2 + 3M\sqrt{\frac{3\ln(T)}{2^{p_r+1}}} + 2M\frac{KM}{(2^{p_r})^2}\right)(22M + 2M\log_2(K))$$

$$\leq M^2 K + (22M + 2M\log_2(K))\left[2MK\log_2(T) + MK\sum_{p_r=0}^{\lceil\log_2 T\rceil}\left(3M\sqrt{\frac{3\ln(T)}{2^{p_r+1}}} + 2\frac{KM^2}{(2^{p_r})^2}\right)\right]$$

$$\leq M^2 K + (22M + 2M\log_2(K))\left[2MK\log_2(T) + MK\left(3M\sqrt{3\ln(T)}\frac{1}{\sqrt{2}-1} + \frac{8KM^2}{3}\right)\right]$$

where inequality (a) is from that there are at most 2 players colliding with each other (leader and one follower) under the nice event $\mathcal{E}_r$. Specifically, with arms in $S_r$ used for communications in epoch $r$,

one communication step leads to a loss at most $2 + V_{\boldsymbol{\mu},*} - V_{\boldsymbol{\mu},S_r}$. Inequality (b) is from that in each epoch $r > 1$, at most $M$ arms statistics need to be communicated. Inequality (c) holds because if the nice event $\mathcal{E}_r$ happens

$$\forall S \in \mathcal{S}_* \backslash \mathcal{S}_c, v(\bar{\boldsymbol{\mu}}_{S_r}^r) \geq v(\bar{\boldsymbol{\mu}}_S^r)$$

$$\Rightarrow \forall S \in \mathcal{S}_* \backslash \mathcal{S}_c, V_{\boldsymbol{\mu},S_r} + M\left(2\sqrt{\frac{3\ln t_r}{2^{p_r+1}}} + \sqrt{\frac{1}{2^{p_r}}}\right) \geq v(\bar{\boldsymbol{\mu}}_{S_r}^r) \geq v(\bar{\boldsymbol{\mu}}_S^r) > v(\boldsymbol{\mu}_S) = V_{\boldsymbol{\mu},*}$$

$$\Rightarrow V_{\boldsymbol{\mu},*} - V_{\boldsymbol{\mu},S_r} \leq M\left(2\sqrt{\frac{3\ln t_r}{2^{p_r+1}}} + \sqrt{\frac{1}{2^{p_r}}}\right) \leq 3M\sqrt{\frac{3\ln(T)}{2^{p_r+1}}};$$

otherwise, the nice event does not happen with $\mathbb{P}(\bar{\mathcal{E}}_r) \leq \frac{2KM}{(2^{p_r})^2}$ proved in the Eqn. (12), $\mathbb{E}[M\mathbb{1}\{\bar{\mathcal{E}}_r\}] \leq 2M\frac{KM}{(2^{p_r})^2}$. $\square$

### F.2 Exploration Regret

**Lemma 5.** *For BEACON, under time horizon $T$, the exploration loss $R_{e,linear}(T)$ is upper bounded as*

$$R_{e,\text{linear}}(T) \leq \sum_{(k,m)} \frac{3727M}{\Delta_{\min}^{k,m}} \ln(T) + 4K^2M^2\Delta_{\max}.$$

*Proof for Lemma 5.* The following proof is based on the proof for CUCB with a linear reward function in Kveton et al. (2015c), but is carefully designed for the complicated batched exploration. In the following proof, we introduce the following notations:

$$S^* = [s_1^*, ..., s_M^*] \in \mathcal{S}_* \backslash \mathcal{S}_c: \text{one particular collision-free optimal matching;}$$
$$\Delta_{S_r} := V_{\boldsymbol{\mu},*} - V_{\boldsymbol{\mu},S_r};$$
$$[\tilde{M}_r] := \{m | m \in [M], s_m^r \neq s_m^*\}.$$

**Step I: Regret decomposition.** First, we can decompose the exploration regret $R_{e,\text{linear}}(T)$ as

$$R_{e,\text{linear}}(T) = \mathbb{E}\left[\sum_r 2^{p_r}(V_{\boldsymbol{\mu},*} - V_{\boldsymbol{\mu},S_r})\right]$$

$$= \mathbb{E}\left[\sum_r 2^{p_r}\Delta_{S_r}\mathbb{1}\{\mathcal{E}_r, \Delta_{S_r} > 0\}\right] + \mathbb{E}\left[\sum_r 2^{p_r}\Delta_{S_r}\mathbb{1}\{\bar{\mathcal{E}}_r, \Delta_{S_r} > 0\}\right] \quad (14)$$

$$\overset{(a)}{\leq} \underbrace{\mathbb{E}\left[\sum_r 2^{p_r}\Delta_{S_r}\mathbb{1}\left\{\sum_{m\in[\tilde{M}_r]}\left(2\sqrt{\frac{3\ln t_r}{2^{p_{s_m^r,m}^r+1}}} + \sqrt{\frac{1}{2^{p_{s_m^r,m}^r}}}\right) \geq \Delta_{S_r}, \Delta_{S_r} > 0\right\}\right]}_{\text{term (C)}}$$

$$+ \underbrace{\mathbb{E}\left[\sum_r 2^{p_r}\Delta_{S_r}\mathbb{1}\{\bar{\mathcal{E}}_r\}\right]}_{\text{term (D)}},$$

where inequality (a) is because when the nice event $\mathcal{E}_r$ happens, choosing a sub-optimal matching $S_r$, i.e., $\Delta_{S_r} > 0$, implies

$$\forall S \in \mathcal{S}_*, v(\bar{\boldsymbol{\mu}}_{S_r}^r) \geq v(\bar{\boldsymbol{\mu}}_S^r)$$
$$\Rightarrow v(\bar{\boldsymbol{\mu}}_{S_r}^r) \geq v(\bar{\boldsymbol{\mu}}_{S^*}^r)$$
$$\Rightarrow \sum_{m\in[\tilde{M}_r]} \bar{\mu}_{s_m^r,m}^r \geq \sum_{m\in[\tilde{M}_r]} \bar{\mu}_{s_m^*,m}^r$$
$$\Rightarrow \sum_{m\in[\tilde{M}_r]} \mu_{s_m^r,m} + \sum_{m\in[\tilde{M}_r]}\left(2\sqrt{\frac{3\ln t_r}{2^{p_{s_m^r,m}^r+1}}} + \sqrt{\frac{1}{2^{p_{s_m^r,m}^r}}}\right) \geq \sum_{m\in[\tilde{M}_r]} \mu_{s_m^*,m}$$

$$\Rightarrow \sum_{m \in [\tilde{M}_r]} \left( 2\sqrt{\frac{3 \ln t_r}{2^{p^r_{s^r_m,m}+1}}} + \sqrt{\frac{1}{2^{p^r_{s^r_m,m}}}} \right) \geq V_{\boldsymbol{\mu},*} - V_{\boldsymbol{\mu},S_r} = \Delta_{S_r}.$$

**Step II: Bounding term (D).** With essentially the same approach of bounding term (B) in the proof of Lemma 3, especially Eqn. (12), we can directly bound term (D) as

$$\text{term (D)} = \mathbb{E}\left[ \sum_r 2^{p_r} \Delta_{S_r} \mathbb{1}\left\{ \bar{\mathcal{E}}_r \right\} \right] \leq 4K^2 M^2 \Delta_{\max}.$$

**Step III: Bounding term (C).** First, we denote event

$$\mathcal{F}_r = \left\{ \sum_{m \in [\tilde{M}_r]} \left( 2\sqrt{\frac{3 \ln t_r}{2^{p^r_{s^r_m,m}+1}}} + \sqrt{\frac{1}{2^{p^r_{s^r_m,m}}}} \right) \geq \Delta_{S_r}, \Delta_{S_r} > 0 \right\},$$

thus

$$\text{term (C)} = \mathbb{E}\left[ \sum_r 2^{p_r} \Delta_{S_r} \mathbb{1}\left\{ \sum_{m \in [\tilde{M}_r]} \left( 2\sqrt{\frac{3 \ln t_r}{2^{p^r_{s^r_m,m}+1}}} + \sqrt{\frac{1}{2^{p^r_{s^r_m,m}}}} \right) \geq \Delta_{S_r}, \Delta_{S^r} > 0 \right\} \right]$$

$$= \mathbb{E}\left[ \sum_r 2^{p_r} \Delta_{S_r} \mathbb{1}\left\{ \mathcal{F}_r \right\} \right].$$

Following the ideas in Kveton et al. (2015c), we introduce two decreasing sequences of constants:

$$1 = b_0 > b_1 > b_2 > \cdots > b_i > \cdots$$
$$a_1 > a_2 > \cdots > a_i > \cdots$$

such that $\lim_{i \to \infty} a_i = \lim_{i \to \infty} b_i = 0$. Furthermore, we specify $q_{i,S_r}$ as the integer satisfying

$$2^{q_{i,S_r}-1} \leq a_i \frac{M^2}{(\Delta_{S_r})^2} \ln(T) < 2^{q_{i,S_r}} \leq 2a_i \frac{M^2}{(\Delta_{S_r})^2} \ln(T).$$

For convenience, we denote $q_{0,S_r} = 0$ and $q_{\infty,S_r} = \infty$. Also, set $H^r_i$ is defined as

$$\forall i \geq 1, H^r_i = \left\{ m \,|\, m \in [\tilde{M}_r], p^r_{s^r_m,m} < q_{i,S_r} \right\},$$

which represents the arms that are not sufficiently sampled compared with $q_{i,S_r}$, and $H^r_0 := [\tilde{M}_r]$.

With the above introduce notations, we define the following infinitely-many events at epoch $r$ as

$G^r_1 = \{ |H^r_1| \geq b_1 M \}$;
$G^r_2 = \{ |H^r_1| < b_1 M \} \cap \{ |H^r_2| \geq b_2 M \}$;
$\cdots$
$G^r_i = \{ |H^r_1| < b_1 M \} \cap \{ |H^r_2| < b_2 M \} \cap \cdots \cap \{ |H^r_{i-1}| < b_{i-1} M \} \cap \{ |H^r_i| \geq b_i M \}$;
$\cdots$

Clearly, these events are mutually exclusive. We have the following proposition.

**Proposition 1.** *Let*

$$\sqrt{14} \sum_{i=1}^{\infty} \frac{b_{i-1} - b_i}{\sqrt{a_i}} \leq 1. \tag{15}$$

*If event $\mathcal{F}_r$ happens at epoch $r$, then there exists $i$ such that $G^r_i$ happens.*

This proposition can be proved by assuming that $\mathcal{F}_r$ happens while none of $G^r_i$ happens. Denoting $\bar{G}_r = \overline{\cup_i G^r_i}$, we can get

$$\bar{G}_r = \overline{\cup_{i=1}^{\infty} G^r_i}$$

$$= \cap_{i=1}^{\infty} \bar{G}_i^r$$
$$= \cap_{i=1}^{\infty} \left[ \left( \cap_{j=1}^{i-1} \left\{ |H_j^r| < b_j M \right\} \right) \cup \overline{\{|H_i^r| \ge b_i M\}} \right]$$
$$= \cap_{i=1}^{\infty} \left[ \left( \cup_{j=1}^{i-1} \overline{\{|H_j^r| < b_j M\}} \right) \cup \overline{\{|H_i^r| \ge b_i M\}} \right]$$
$$= \cap_{i=1}^{\infty} \left[ \left( \cup_{j=1}^{i-1} \left\{ |H_j^r| \ge b_j M \right\} \right) \cup \{|H_i^r| < b_i M\} \right]$$
$$= \cap_{i=1}^{\infty} \left\{ |H_i^r| < b_i M \right\}.$$

If $\bar{G}_r$ happens, denoting $\tilde{H}_i^r = [\tilde{M}_r] \backslash H_i^r$, which implies $\tilde{H}_{i-1}^r \subseteq \tilde{H}_i^r$ and $[\tilde{M}_r] = \cup_i (\tilde{H}_i^r \backslash \tilde{H}_{i-1}^r)$, then it holds that

$$\sum_{m \in [\tilde{M}_r]} \left( 2\sqrt{\frac{3 \ln T}{2^{p_{s_m^r,m}^r + 1}}} + \sqrt{\frac{1}{2^{p_{s_m^r,m}^r}}} \right)$$

$$\le 3\sqrt{3 \ln T} \sum_{m \in [\tilde{M}_r]} \frac{1}{\sqrt{2^{p_{s_m^r,m}^r + 1}}}$$

$$= 3\sqrt{3 \ln T} \sum_{i=1}^{\infty} \sum_{m \in \tilde{H}_i^r \backslash \tilde{H}_{i-1}^r} \frac{1}{\sqrt{2^{p_{s_m^r,m}^r + 1}}}$$

$$= 3\sqrt{3 \ln T} \sum_{i=1}^{\infty} \frac{|\tilde{H}_i^r \backslash \tilde{H}_{i-1}^r|}{\sqrt{2^{q_{i-1,S_r} + 1}}}$$

$$\le 3\sqrt{3 \ln T} \sum_{i=1}^{\infty} \frac{|\tilde{H}_i^r \backslash \tilde{H}_{i-1}^r|}{\sqrt{2 a_i \frac{M^2}{(\Delta_{S_r})^2} \ln(T)}}$$

$$\le 3\sqrt{3/2} \frac{\Delta_{S_r}}{M} \sum_{i=1}^{\infty} \left( |H_{i-1}^r| - |H_i^r| \right) \frac{1}{\sqrt{a_i}}$$

$$= 3\sqrt{3/2} \frac{\Delta_{S_r}}{M} |H_0^r| \frac{1}{\sqrt{a_1}} + 3\sqrt{3/2} \frac{\Delta_{S_r}}{M} \sum_{i=1}^{\infty} |H_i^r| \left( \frac{1}{\sqrt{a_{i+1}}} - \frac{1}{\sqrt{a_i}} \right)$$

$$\overset{(a)}{\le} 3\sqrt{3/2} \frac{\Delta_{S_r}}{M} b_0 M \frac{1}{\sqrt{a_1}} + 3\sqrt{3/2} \frac{\Delta_{S_r}}{M} \sum_{i=1}^{\infty} b_i M \left( \frac{1}{\sqrt{a_{i+1}}} - \frac{1}{\sqrt{a_i}} \right)$$

$$< \sqrt{14} \sum_{i=1}^{\infty} \frac{b_{i-1} - b_i}{\sqrt{a_i}} \Delta_{S_r}$$

$$\le \Delta_{S_r},$$

where inequality is because $|H_i^r| < b_i M$ with $\bar{G}_r$ happening. This result contradicts with the definition of $\mathcal{F}_r$ as

$$\mathcal{F}_r = \left\{ \sum_{m \in [\tilde{M}^r]} \left( 2\sqrt{\frac{3 \ln t_r}{2^{p_{s_m^r,m}^r + 1}}} + \sqrt{\frac{1}{2^{p_{s_m^r,m}^r}}} \right) \ge \Delta_{S_r}, \Delta_{S_r} > 0 \right\}.$$

With Proposition 1, when Eqn. (15) holds, we can further decompose term (C) as

$$\text{term (C)} = \mathbb{E} \left[ \sum_r 2^{p_r} \Delta_{S_r} \mathbb{1} \left\{ \mathcal{F}_r \right\} \right] = \mathbb{E} \left[ \sum_r \sum_{i=1}^{\infty} 2^{p_r} \Delta_{S_r} \mathbb{1} \left\{ G_i^r, \Delta_{S_r} > 0 \right\} \right].$$

Then, the following events are defined

$$G_{i,k,m}^r = G_i^r \cap \left\{ m \in [\tilde{M}_r], s_m^r = k, p_{k,m}^r < q_{i,S_r} \right\},$$

which imply that

$$\mathbb{1} \left\{ G_i^r, \Delta_{S_r} > 0 \right\} \le \frac{1}{b_i M} \sum_{(k,m)} \mathbb{1} \left\{ G_{i,s_m^r,m}^r, \Delta_{S_r} > 0 \right\}$$

since at least $b_i M$ arms with event $G^r_{i,k,m}$ happening are required to make $G^r_i$ happen.

Thus, recall $\mathcal{S}^{k,m}_b = \{S | S \in \mathcal{S}_b, s_m = k\} = \{S^{k,m}_1, ..., S^{k,m}_{N(k,m)}\}$, we can get

$$
\text{term (C)} = \mathbb{E}\left[\sum_r \sum_{i=1}^{\infty} 2^{p_r} \Delta_{S_r} \mathbb{1}\left\{G^r_i, \Delta_{S_r} > 0\right\}\right]
$$

$$
\leq \mathbb{E}\left[\sum_r \sum_{i=1}^{\infty} 2^{p_r} \Delta_{S_r} \frac{1}{b_i M} \sum_{(k,m)} \mathbb{1}\left\{G^r_{i,k,m}, \Delta_{S_r} > 0\right\}\right]
$$

$$
\leq \mathbb{E}\left[\sum_r \sum_{i=1}^{\infty} 2^{p_r} \Delta_{S_r} \frac{1}{b_i M} \sum_{(k,m)} \mathbb{1}\left\{m \in [\tilde{M}_r], s^r_m = k, p^r_{k,m} < q_{i,S_r}, \Delta_{S_r} > 0\right\}\right]
$$

$$
= \mathbb{E}\left[\sum_{(k,m)} \sum_{n=1}^{N(k,m)} \sum_r \sum_{i=1}^{\infty} 2^{p_r} \frac{1}{b_i M} \mathbb{1}\left\{s^r_m = k, p^r_{k,m} < q_{i,S^{k,m}_n}, S_r = S^{k,m}_n\right\} \Delta^{k,m}_n\right]
$$

$$
= \mathbb{E}\left[\sum_{(k,m)} \sum_{i=1}^{\infty} \underbrace{\sum_r \sum_{n=1}^{N(k,m)} 2^{p_r} \frac{1}{b_i M} \mathbb{1}\left\{s^r_m = k, p^r_{k,m} < q_{i,S^{k,m}_n}, S_r = S^{k,m}_n\right\} \Delta^{k,m}_n}_{\text{term (E)}}\right]
$$

$$
\overset{(a)}{\leq} \mathbb{E}\left[\sum_{(k,m)} \left[\sum_{i=1}^{\infty} \frac{6 a_i}{b_i}\right] \frac{M}{\Delta^{k,m}_{N(k,m)}} \ln(T)\right]
$$

where inequality (a) holds because term (E) can be bounded as

$$
\text{term (E)} = \sum_r \sum_{n=1}^{N(k,m)} 2^{p_r} \frac{1}{b_i M} \mathbb{1}\left\{s^r_m = k, p^r_{k,m} < q_{i,S^{k,m}_n}, S_r = S^{k,m}_n\right\} \Delta^{k,m}_n
$$

$$
\leq 3 \times 2^{q_{i,S^{k,m}_1} - 1} \frac{\Delta^{k,m}_1}{b_i M} + \frac{1}{b_i M} \sum_{n=2}^{N(k,m)} \left(3 \times 2^{q_{i,S^{k,m}_n} - 1} - 3 \times 2^{q_{i,S^{k,m}_{n-1}} - 1}\right) \Delta^{k,m}_n
$$

$$
\leq \frac{3 a_i M}{b_i \Delta^{k,m}_1} \ln(T) + \frac{3 a_i M}{b_i} \sum_{n=2}^{N(k,m)} \left(\frac{1}{(\Delta^{k,m}_n)^2} - \frac{1}{(\Delta^{k,m}_{n-1})^2}\right) \Delta^{k,m}_n \ln(T)
$$

$$
= \frac{3 a_i M}{b_i} \ln(T) \left[\sum_{n=1}^{N(k,m)-1} \frac{\Delta^{k,m}_n - \Delta^{k,m}_{n+1}}{(\Delta^{k,m}_n)^2} + \frac{1}{\Delta^{k,m}_{N(k,m)}}\right]
$$

$$
\leq \frac{3 a_i M}{b_i} \ln(T) \left[\sum_{n=1}^{N(k,m)-1} \frac{\Delta^{k,m}_n - \Delta^{k,m}_{n+1}}{\Delta^{k,m}_n \Delta^{k,m}_{n+1}} + \frac{1}{\Delta^{k,m}_{N(k,m)}}\right]
$$

$$
\leq \frac{3 a_i M}{b_i} \ln(T) \frac{2}{\Delta^{k,m}_{N(k,m)}}.
$$

At last, we specify the choices of $a_i$ and $b_i$, which resolve to the following optimization problem:

minimize $\displaystyle\sum_{i=1}^{\infty} \frac{6 a_i}{b_i}$

subject to $\displaystyle\lim_{i\to\infty} a_i = \lim_{i\to\infty} b_i = 0$

Monotonicity: $1 = b_0 > b_1 > b_2 > \cdots > b_i > \cdots ; a_1 > a_2 > \cdots > a_i > \cdots$

Eqn. (15): $\displaystyle\sqrt{14} \sum_{i=1}^{\infty} \frac{b_{i-1} - b_i}{\sqrt{a_i}} \leq 1.$

We choose $a_i$ and $b_i$ to be geometric sequences as in Kveton et al. (2015c), specifically $a_i = d(a)^i$ and $b_i = (b)^i$ with $0 < a, b < 1$ and $d > 0$. Moreover, if $b \leq \sqrt{a}$, to meed Eqn. (15), it needs

$$\sqrt{14} \sum_{i=1}^{\infty} \frac{b_{i-1} - b_i}{\sqrt{a_i}} = \sqrt{14} \sum_{i=1}^{\infty} \frac{(b)^{i-1} - (b)^i}{\sqrt{d(a)^i}} = \sqrt{\frac{14}{d}} \frac{1-b}{\sqrt{a} - b} \leq 1 \Rightarrow d \geq 14 \left( \frac{1-b}{\sqrt{a} - b} \right)^2.$$

Thus, the best choice for $d$ is $d = 14 \left( \frac{1-b}{\sqrt{a}-b} \right)^2$ and the problem is reformulated as

$$\text{minimize} \sum_{i=1}^{\infty} \frac{6a_i}{b_i} = 84 \left( \frac{1-b}{\sqrt{a}-b} \right)^2 \frac{\alpha}{b-a}$$

$$\text{conditioned on } 0 < a < b < \sqrt{a} < 1.$$

With numerically calculated $a = 0.1459$ and $b = 0.2360$ in Kveton et al. (2015c), we get $\sum_{i=1}^{\infty} \frac{6a_i}{b_i} \leq 3727$. Thus, we conclude that

$$\text{term (C)} \leq \mathbb{E} \left[ \sum_{(k,m)} \left[ \sum_{i=1}^{\infty} \frac{6a_i}{b_i} \right] \frac{M}{\Delta_{N(k,m)}^{k,m}} \ln(T) \right]$$

$$\leq \sum_{(k,m)} \frac{3727M}{\Delta_{N(k,m)}^{k,m}} \ln(T)$$

$$\leq \sum_{(k,m)} \frac{3727M}{\Delta_{\min}^{k,m}} \ln(T).$$

Lemma 5 can be proved by combining term (C) and term (D). $\qquad \square$

## G  Proof for Theorem 2

*Proof.* This proof follows naturally from Theorem 5 by categorizing sub-optimal gaps with a threshold $\epsilon$.

Specifically, we can modify Eqn. (14) as

$$R_{e,\text{linear}}(T) = \mathbb{E} \left[ \sum_r 2^{p_r} (V_{\boldsymbol{\mu},*} - V_{\boldsymbol{\mu}, S_r}) \right]$$

$$= \mathbb{E} \left[ \sum_r 2^{p_r} \Delta_{S_r} \mathbb{1} \{\mathcal{E}_r, \Delta_{S_r} > 0\} \right] + \mathbb{E} \left[ \sum_r 2^{p_r} \Delta_{S_r} \mathbb{1} \{\bar{\mathcal{E}}_r, \Delta_{S_r} > 0\} \right]$$

$$\leq T\epsilon + \mathbb{E} \left[ \sum_r 2^{p_r} \Delta_{S_r} \mathbb{1} \{\mathcal{E}_r, \Delta_{S_r} > \epsilon\} \right] + \mathbb{E} \left[ \sum_r 2^{p_r} \Delta_{S_r} \mathbb{1} \{\bar{\mathcal{E}}_r, \Delta_{S_r} > \epsilon\} \right]$$

$$\overset{(a)}{\leq} T\epsilon + \sum_{(k,m)} \frac{3727M}{\epsilon} \ln(T) + 4K^2 M^2 \Delta_{\max},$$

where inequality (a) follows the same proof for Lemma 5. For the overall regret, we can further get

$$R_{\text{linear}}(T) \leq T\epsilon + \frac{3727M^2 K}{\epsilon} \ln(T) + \text{terms of order } O(\ln(T)) \text{ and independent with } \epsilon$$

$$\overset{(a)}{\leq} 124M\sqrt{KT\ln(T)} + \text{terms of order } O(\ln(T)) \text{ and independent with } \epsilon$$

$$= O\left( M\sqrt{KT\log(T)} \right),$$

where $\epsilon$ is taken as $62M\sqrt{\frac{K\ln(T)}{T}}$ in inequality (a). Theorem 2 is then proved. $\qquad \square$

# H    $(\alpha, \beta)$-**Approximation Oracle and Regret**

In this section, we discuss how to extend from exact oracles to $(\alpha, \beta)$-approximation oracles, and the corresponding performance guarantees. With the definition given in Section 5.2, it is straightforward to use $(\alpha, \beta)$-approximation oracles to replace the original exact oracles in BEACON. To facilitate the discussion, we further assume that this approximation oracle always outputs collision-free matchings, which naturally holds for most of approximate optimization solvers (Vazirani, 2013).

With an $(\alpha, \beta)$-approximation oracle, as stated in Section 5.2, a regret bound similar to Theorem 3 can be obtained regarding the $(\alpha, \beta)$-approximation regret. First, the following notations are redefined and slightly abused to accommodate the $(\alpha, \beta)$-approximation regret: $\mathcal{S}_* = \{S | S \in \mathcal{S}, V_{\boldsymbol{\mu},S} \geq \alpha V_{\boldsymbol{\mu},*}\}$: the set of matchings with rewards larger than $\alpha V_{\boldsymbol{\mu},*}$; $\Delta_{\min}^{k,m} = \alpha V_{\boldsymbol{\mu},*} - \max\{V_{\boldsymbol{\mu},S} | S \in \mathcal{S}_b, s_m = k\}$; $\Delta_{\max}^{k,m} = \alpha V_{\boldsymbol{\mu},*} - \min\{V_{\boldsymbol{\mu},S} | S \in \mathcal{S}_b, s_m = k\}$. With these notations, BEACON's performance with an approximate oracle is established in the following.

**Theorem 6** ($(\alpha, \beta)$-approximation regret). *Under Assumptions 1, 2, and 3, with an $(\alpha, \beta)$-approximation oracle, the $(\alpha, \beta)$-approximation regret of BEACON is upper bounded as*

$$R(T) = \tilde{O}\left(\sum_{(k,m) \in [K] \times [M]}\left[\frac{\Delta_{\min}^{k,m}}{(f^{-1}(\Delta_{\min}^{k,m}))^2} + \int_{\Delta_{\min}^{k,m}}^{\Delta_{\max}^{k,m}} \frac{1}{(f^{-1}(x))^2}\mathrm{d}x\right]\log(T) + M^2 K \Delta_c \log(T)\right).$$

*Proof.* The proof for Theorem 6 closely follows the proof for Theorem 3. To avoid unnecessarily redundant exposition, we here only highlight the key steps and major differences.

The communication regret and the other regret can be obtained with the same approach in the proof for Theorem 3. The main difference lies in the exploration regret. In the following proof, unless specified explicitly before, the adopted notations share the same definition as in the proof for Theorem 3. Similar to Eqn. (11), we can decompose the exploration regret w.r.t. the definition of the $(\alpha, \beta)$-approximation regret as

$$
\begin{aligned}
R_e(T) &= \mathbb{E}\left[\sum_r 2^{p_r}(\alpha\beta V_{\boldsymbol{\mu},*} - V_{\boldsymbol{\mu},S_r})\right] \\
&= \mathbb{E}\left[\sum_r 2^{p_r}(\alpha V_{\boldsymbol{\mu},*} - V_{\boldsymbol{\mu},S_r})\right] + \alpha(\beta - 1)V_{\boldsymbol{\mu},*}\mathbb{E}[T_e] \\
&= \mathbb{E}\left[\sum_r 2^{p_r}(\alpha V_{\boldsymbol{\mu},*} - V_{\boldsymbol{\mu},S_r})(\mathbb{1}\{\mathcal{G}_r\} + \mathbb{1}\{\bar{\mathcal{G}}_r\})\right] + \alpha(\beta - 1)V_{\boldsymbol{\mu},*}T_e \\
&\leq \mathbb{E}\left[\sum_r 2^{p_r}(\alpha V_{\boldsymbol{\mu},*} - V_{\boldsymbol{\mu},S_r})(\mathbb{1}\{\mathcal{G}_r\} + 1 - \beta)\right] + \alpha(\beta - 1)V_{\boldsymbol{\mu},*}T_e \\
&\leq \mathbb{E}\left[\sum_r 2^{p_r}(\alpha V_{\boldsymbol{\mu},*} - V_{\boldsymbol{\mu},S_r})\mathbb{1}\{\mathcal{G}_r\}\right]
\end{aligned}
$$

where $T_e$ is the length of overall exploration phases. Notation $\mathcal{G}_r := \{V_{\boldsymbol{\mu},S_r} \geq \alpha V_{\boldsymbol{\mu},*}\}$ denotes the event that the oracle successfully outputs a good matching at epoch $r$, which happens with a probability at least $\beta$. Then, conditioned on event $\mathcal{G}_r$, the remaining analysis follows the same process in the proof for Lemma 3, and Theorem 6 can be obtained. $\square$