# OpenReview forum: "Heterogeneous Multi-player Multi-armed Bandits: Closing the Gap and Generalization"
_NeurIPS.cc/2021/Conference — NeurIPS 2021 Poster_

### Official Review · Reviewer_Vyvg · 2021-07-14

**Rating:** 6
**Confidence:** 3

**Summary:**

Overall, this paper introduces a new algorithm BEACON for multi-agent multi-armed bandits that closes the gap between the state-of-the-art upper bound and information theoretic lower bound of the regrets by implicit communication via adaptive differential communication protocols and efficient exploration via UCB-based algorithms. In addition to the existing case where the system-level reward is a linear combination of individual rewards, BEACON also tackles the general cases where the combination is non-linear, obtaining a regret bound of O(log T). Experiments show that BEACON significantly outperforms state-of-the-art benchmarks in decentralized bandits.

**Ethical Concerns:**

Not applicable.

**Limitations And Societal Impact:**

Limitations:  BEACON depends on an oracle solver for combinatorial optimization, a requirement of which might make BEACON computationally infeasible when the numbers of arms and players becomes extremely large.

Societal Impact: Not applicable. The paper is largely algorithmic and centers on theoretical justification of an abstract model. If any, the potential impact of this paper to the research community would be beneficial as it provides new methods for cooperative bandit decision making.


**Main Review:**

Strengths:
1. The paper is well-written, with all the sections clearly explained and the contributions clearly stated. The authors have made comprehensive comparisons against state-of-the-art benchmarks from prior work, and have clearly stated future directions from the work.

2. The contributions of this paper to the research community are evident and non-trivial. Using batch processing along with implicit communication protocols, BEACON not only closes the regret gap to information-theoretic lower bounds, but also generalizes to general reward functions with fewer assumptions/prerequisites.

3. Proper experiments have been conducted, with the necessary comparisons with existing state-of-the-art bandit algorithms included in the paper.

Weaknesses:
1. Rather than putting all discussion of the connection between Combinatorial MAB and MP-MAB into the supplementary materials, it would be beneficial to provide a summary as to how MP-MAB could be reduced to/relates to a CMAB problem. What does ‘the largest amount of matchings’ stand for in Section 3.1? The descriptions at this point are inadequate and don’t provide enough intuition.

2. It is not immediately clear why the differential bit representation in the ADC protocol would help reduce the overall (expected) regret at each communication epoch to O(1). It would be ideal to include a brief description(at least intuitively) how truncating bits of the intra-epoch mean difference would result in regret reduction.

3. Are there any constraints with respect to the number of players each player is able to communicate with? It appears from the paper that BEACON sets constraints on the number of times a player(as leader) communicates with other players(as followers ) during communication, but not who to communicate with. What does the underlying communication graph amongst players look like? Some additional clarification would be helpful for further understanding.

Overall, my recommendation would be a ‘6, marginally above’. I’d be happy to revise my score, if the authors address my concerns accordingly.


**Time Spent Reviewing:**

6

---

> ### Author Response · Authors · 2021-08-10
> **Response to Reviewer Vyvg**
>
> We thank the reviewer for the valuable suggestions and comments. Below we address the reviewer's comments in order.
>
> - We agree with the reviewer that it is beneficial to have some discussions in the main paper to connect MP-MAB and CMAB, which will be added to the revised main paper. Here, we briefly note that MP-MAB can be thought of as a decentralized version of CMAB, while this decentralized nature leads to additional challenges with collision-avoidance and information sharing.
>
> - In Section 3.1, the "large amount of matching" refers to the fact that in the heterogeneous model, there could be a large number of combinations between players and arms. Rigorously speaking, the set of all possible matchings $\mathcal{S}$ has a large cardinality of $K^M$, and even the set of non-collision matchings has a cardinality of $P(K,M)$. This combinatorial structure requires the designed algorithm to effectively explore among such a large amount of matchings. BEACON nicely resolves this obstacle by the adopted ideas from CUCB, which contributes to its capability of approaching the centralized performance. We will put more discussions on this to the revised paper.
>
> - The most fundamental idea behind ADC is that the (quantized) sample means of each arm in different epochs are all concentrated around their true means. This concentration results in small differences among these sample means, which become even smaller as the bandit game progresses. Then, after truncating, only informative bits in these small differences are kept (i.e., uninformative bit zeros are discarded), which leads to the superior efficiency of ADC. Some illustrations can be found in line 190--196, and we will add more discussions to the revised paper.
>
> - Regarding the reviewer's question on the communication constraint, we would like to clarify with the following three points.
>
>   (I) Due to the nature of forced-collision communication, there is no constraint on whom or how many players that one can implicitly communicate with. Specifically, BEACON performs communications implicitly via (forced) collisions among players and collisions can occur between any two players under the current problem formulation. Thus, the adopted implicit communication can also naturally happen between arbitrary players..
>
>   (II) The constraint on the times of implicit communications is from the goal of algorithm design (instead of the problem formulation). Specifically, since each bit of transmission with implicit communication costs at least $O(1)$ due to collisions, the designed algorithm should only communicate $O(\log(T))$ times to have a $O(\log(T))$ regret, which is the main difficulty in communication protocol design. Our proposed ADC protocol nicely overcomes this obstacle with a highly efficient communication scheme.
>
>   (III) We believe with points (I) and (II), it would be clear that there is no explicit underlying communication graph from the problem formulation. As an algorithmic choice, BEACON adopts the leader-follower structure for the designed implicit communication, which can effectively aggregate information while lowering the communication cost.
>
> - Regarding the reviewer's concern on the combinatorial optimization oracle, as stated in line 315--321, we agree that BEACON does rely on it and this can be an interesting direction for future works. However, we also note that for linear reward functions, this optimization can be solved with polynomial time complexity as stated in line  138. Furthermore, for general reward functions, BEACON's compatibility with the $(\alpha,\beta)$-approximate oracle would also ease this computation burden as stated in line 282-287.

---

### Official Review · Reviewer_CoDr · 2021-07-15

**Rating:** 7
**Confidence:** 5

**Summary:**

This paper studies heterogeneous multiplayer bandits (with collision sensing). Designing a new communication protocol, it proposes a batched and decentralized version of CUCB to reach centralized performances in heterogeneous multiplayer bandits.
The proposed algorithm can also be extended to more general reward functions, similarly to CUCB in combinatorial bandits.

**Limitations And Societal Impact:**

Not concerned

**Main Review:**

I have read the other reviews and the authors' answers and maintain my positive opinion about this work.

-----------------------------------------

I find this work very interesting: nicely combining combinatorial bandits and multiplayer bandits was necessary to reach near centralized bound in the heterogeneous setting and is done nicely here, thanks to the differential communication protocol (which leads to log(T) communication rounds instead of log^2(T)).

The paper is well written, although we feel that the authors struggled to fit the 9 pages limit (eg it contains many footnotes): the description of the algorithm is very clear and the analysis seems to follow from the analysis of CUCB once the communication protocols are carefully handled. I have no major concern regarding this work, hence my overall score.

- Note that if the horizon T is a priori known, using epochs of length 2^p log(T) (instead of 2^p) leads to a constant (in T) number of communication phases for existing ETC algorithms. Could such an improvement be combined with differential communication to decrease even further the communication cost? This could be of great interest, especially without sensing information, as sending a single bit here has a cost of order log(T)

------- minor comments --------

- I think the "matching" denomination is poorly chosen, as matching usually means that no arm is assigned to several players (ie it corresponds to the set S_b here)

- line 83: V_{mu, S} is not well defined in case of collisions. It is here suggested that collisions do not degrade the total reward.

- The length of communication is random (and thus unknown) to the different players. This is an additional challenge of the algorithm that is only mentioned in the Appendix. If the authors manage to save some room, I suggest to at least mention this additional difficulty in Section 3.3

- equation (3): in the last formulation, the second term is not necessary since it is dominated by the first one

- in table 1: I think it would be better to remove the tilde on O for all the algorithms that do not incur extra log(K) terms

- Section 6 does not really seem at its right place in the current structure. I would rather postpone it after the experiments (or before the conclusion)

**Time Spent Reviewing:**

4 h

---

> ### Author Response · Authors · 2021-08-10
> **Response to Reviewer CoDr**
>
> We thank the reviewer for the thoughtful suggestions and comments.
>
> - The reviewer's suggestion about combining the differential communication and existing ETC schemes is brilliant, and this is exactly where our initial thoughts of ADC came from. We are happy to elaborate on this aspect as follows. We will also add the discussions to the revised paper.
>
>   (I) It is indeed feasible to combine ADC and existing ETC schemes, e.g., METC (Boursier et al., 2020), and if there is exactly one optimal matching, this combination does help reduce the communication cost to a constant with the suggested epoch length. Also, as the reviewer correctly noted, this result can be extended to no-sensing settings with a communication cost of $O(\log(T))$.  However, when there are potentially **multiple** optimal matchings (as considered in this work), the communication cost will still be $O(\log(T))$. The reason is that due to the existence of multiple optimal matchings, METC would fail to converge and thus keep communicating until the end of the time horizon, which results in at most $O(\log(T))$ epochs.
>
>   (II) We also note that, with ideas from CUCB, BEACON is much more efficient in exploration than ETC-type of algorithms (e.g., METC), which is the main reason we did not fully elaborate on the suggested combination of ETC and ADC in this paper. Theoretically, this superiority can be reflected in the extra multiplicative factor $M$ in the exploration loss of METC shown in Table 1.
>
> - We sincerely thank the reviewer for the kind comments and suggestions regarding our improperly chosen denomination, careless typo with $V_{\mu,S}$ and beyond. We will address them accordingly in the revised paper.

---

### Official Review · Reviewer_5RkM · 2021-07-15

**Rating:** 6
**Confidence:** 3

**Summary:**

This paper proposes the BEACON algorithm under Multi-player MAB settings. The authors make use of the ideas in combinatorial-MAB area and implement batched exploration, in which they choose an update rule for the arm counters to sufficiently but not excessively explore the arms. Then, they  extends the adaptive communication to adaptive differential communication, also, the communication only happens when the sample mean is sufficiently precise. With the combination of these strategies, the BEACON achieves the regret lower bound while requiring less assumptions (unknown horizon, non-linear reward functions, etc.) than prior works in the area.

**Limitations And Societal Impact:**

As the author pointed out, the work utilizes a combinatorial optimization ORACLE, which can be infeasible when the scale is large. There aren’t any negative societal impact of their work that I can identify.

**Main Review:**

Originality: The combination of implicit communication for distributed operation with the use of centralized UCB (CUCB) seems like the main novelty of the work. The idea of using differential encoding to reduce the communication load, while natural, is useful.

Clarity: The paper is very clearly written and well organized, even though the notation is heavy, the clear presentation make it sufficiently easy to read and understand the paper. The authors explain the key ideas very clearly. The literature review appear sufficient as well. In particular, the related works are carefully discussed and compared to emphasize the contributions.

Significance: The proposed BEACON achieves the regret lower bound while requiring less assumptions than earlier works. Also, the possibility of allowing the reward function to be non-linear may be significant. While these technical results appear solid, I am not convinced in the importance of the problem setting. The need for implicit communication amongst distributed agents via collisions is quite artificial and forced. I do not see strong justification for enforcing such a constraint especially in a cooperative setting such as the one in this work. It may be more acceptable is a non-cooperative setting where users have self-interests. However, this would make the existing design useless, as it assumes thrust.

Minor question: why is the expected system reward V_{\mu,S} given by the sum_m \mu_{s_m,m} in line 84? Shouldn't there be an expectation over the  no-collision indicator?


**Time Spent Reviewing:**

2 hours

---

> ### Author Response · Authors · 2021-08-10
> **Response to Reviewer 5RkM**
>
> We thank the reviewer for the detailed comments.
>
> - Regarding the reviewer's question on the importance of the problem setting, we would like to clarify from the following three aspects.
>
>   (I) The decentralized MP-MAB problem with collisions among players is a well-established model that has been studied for years (see the references listed in Section 8). It has clear real-world motivations from cognitive radio (Liu and Zhao, 2010; Avner and Mannor, 2014), where multiple distributed users have to share communication channels and using the same channel results in collisions. Thus, it is meaningful and practical to consider this problem.
>
>   (II) Under this well-established model, the implicit communication via collisions is an algorithmic choice (instead of a constraint) to promote player interactions, which is one of the major difficulties of the underlying model since explicit information sharing is prohibited. It has been adopted for both the homogeneous and heterogeneous models in many prior works (Boursier and Perchet, 2019; Wang et al., 2020; Shi et al,. 2020; Boursier et al., 2020), even for the adversarial version of MP-MAB (Alatur et al., 2020; Shi and Shen, 2021), and is a well-accepted tool for solving the decentralized MP-MAB problem.
>
>   (III) As the reviewer pointed out, it is indeed interesting to consider the non-cooperative setting, but that does not devalue the efforts in this work. For example, some progress have been made in [R1] to address selfish/malicious players, and implicit communications prove to be still useful with some extensions for robustness. Similarly, we believe the design ideas in this work, especially the ADC protocol, can also be of interest beyond the fully cooperative setting. We will highlight this as a future direction in the revised paper and add the corresponding discussions.
>
>   [R1] Boursier, E., \& Perchet, V. (2020, July). Selfish robustness and equilibria in multi-player bandits. In Conference on Learning Theory (pp. 530-581). PMLR.
>
> - We thank the reviewer for kindly pointing out our careless typo with the expected system reward, which should indeed contain the no-collision indicator, i.e., $V_{\mu,S} = \sum_m \mu_{s_m,m}\eta_{s_m}(S)$. We will fix it in the revised paper.

---

### Official Review · Reviewer_5qQj · 2021-07-22

**Rating:** 5
**Confidence:** 3

**Summary:**

The paper studies the decentralized multi-armed bandit problem and proposes an algorithm, appealing to the ideas of batched exploration and adaptive communication, which has the same order of regret as the centralized version of the problem.

**Limitations And Societal Impact:**

Yes

**Main Review:**

The decentralized heterogeneous bandit problem is important and hard. Both the ideas of batched exploration and adaptive communication are interesting. My concern of the paper is as follows.

The paper aims to close the gap in regrets between the centralized and decentralized settings. However, the proposed algorithm heavily relies on centralized design. For example, there is a leader among all the agents which can collect all information from all agents and perform a centralized optimization task at each round. Another example is that the batched exploration procedure also depends on a network-wise pieces of information, like the smallest arm counter. With these, it is not reasonable to me to call the algorithm decentralized. More importantly, with allowing such information collection and processing in a "center", it can be expected that the proposed algorithm can perform as good as (or even outperform) the centralized algorithm.

Update Sep 6: Thanks the authors for their responses.

The paper claims to close the "gap" between centralized and decentralized regrets. However, the authors to some extend "mixed" the existing literature. The original decentralized setting does not allow communication among the agents (e.g. Kalathil et al. 2014 and Nayyar et al. 2016 in the Related Works section, as well as the NIPS paper Bistritz and Leshem, 2018 in the authors' response); this is why the gap exists in my understanding. Now, the paper follows some recent development and allows communication among the agents. To me, this significantly changes the game. I would call it a distributed/cooperative setting, since it requires that all the agents coordinate (have to send correct information). In the vast distributed convex optimization (which allows communication among the agents and the communication graph can be sparse, as long as connected) literature, it is known that distributed convex optimization algorithms can achieve the same order of convergence rate as the centralized counterpart; the difference is in the constant coefficient level, which depends on the graph connectivity. Intuitively, the more connected the graph is, the faster (better constant coefficient) the convergence is. Since the current paper makes use of a leader-follower scheme in communication, it essentially corresponds to a complete graph in a distributed setting. So it is not surprising to me that it can achieve the regret close to the centralized setting. Also, from a distributed algorithm design point of view, assigning a unique ID to each agent, as the current paper does, is a very strong assumption; that is why in most distributed optimization algorithms, there does not exist a "leader".

With the above in mind, my understanding of the paper contribution is how to design a communication efficient scheme between the leader and each follower to achieve an "optimal" tradeoff between communication and system regret, instead of closing the "gap" between centralized and decentralized regrets, because it essentially considers a very special distributed setting. Note that transmitting one bit is essentially different from transmitting more bits as the former implicitly requires establishing new communication channels.

From my focus point of view (the distributed/decentralized setting), the paper's scope is limited and the title could be misleading, so I keep my score 5.


**Time Spent Reviewing:**

6

---

> ### Author Response · Authors · 2021-08-10
> **Response to Reviewer 5qQj**
>
> We thank the reviewer for the thoughtful comments, and would like to address the concern as follows.
>
> - We want to first clarify the perception that with a centralized-inspired algorithm, it is relatively easy to approach or even outperform the centralized performance. We would like to point out that our work is the first to show that this is possible -- similar algorithmic choices of implicit communication and network-wise leader-follower information processing were also adopted in several prior works (Boursier et al., 2020; Tibrewal et al., 2019) but none of them were capable of closing the gap to the centralized performance. The ADC design, batched exploration, and the CUCB principle are the novel algorithmic components that collectively close this gap, which is one of the major contributions of this paper.
> - Regarding the centralized-versus-decentralized algorithm design, as the reviewer pointed out, the proposed BEACON algorithm is indeed *inspired* by centralized algorithms, especially CMAB. However, the problem formulation itself **exactly** follows the existing decentralized MP-MAB problem setting (Tibrewal et al., 2019; Bistritz and Leshem, 2018, 2020; Boursier et al., 2020); in this sense, we consider the algorithm decentralized because it solves a fundamentally decentralized MP-MAB problem. For the same problem formulation, it is then the designer's choice of the algorithm structure as long as it satisfies the problem constraint. Adopting a leader-follower architecture does tilt the design towards being more "centralized", but this is again an algorithmic choice that has been adopted in prior literature for decentralized MP-MAB (Boursier et al., 2020; Shi et al., 2020). We have made a conscious attempt to highlight this choice, and suggested that one of the future directions is how to remove the centralized reliance (as stated in line 315--321). At the same time, we believe that the design and analysis of BEACON are valuable and non-trivial, especially the adopted CUCB ideas, the proposed ADC protocol, and the revealed intimate relationship between CMAB and MP-MAB.

---

> ### Author Response · Authors · 2021-09-07
> **Response to updated comment from Reviewer 5qQj**
>
> We thank the reviewer for kindly updating comments. We summarize the reviewer's comments into the following points and would like to make some corresponding clarifications.
>
> - **Point 1 (mixing settings)**: The game is fundamentally changed with implicit communications via collisions and this work mixed the settings.
>
>   **Ans**: We would like to clarify that the settings in MP-MAB are consistent with each other, and our adopted formulation exactly follows the existing standard one. This setting forbids **explicit** communications (via external channels), but does not restrict players to interact with each other via arm collisions. Since the existence of collisions is fundamental, leveraging such collisions to help, as opposed to letting them degrade the performance, is within the considered model. It is then an algorithmic choice whether to use **implicit** communication via collisions or not, instead of a model requirement.
>
>   The following quote from Section V in Kalathil et al. (2014), one of the no-communication references listed by Reviewer 5qQj, may be helpful to verify this point: *"We assume that there are no dedicated control channels for coordination or communication between the players. However, we do allow for players to communicate with each other by pulling arms in a specific order. ... However, such a communication overhead would add to regret and affect the learning rate."*
>
>   More concretely, players need to observe and create collisions in order to coordinately run the Bertsekas auction algorithm in Kalathil et al. (2014) and Nayyar et al. (2016). Similarly, the GoT phase in Bistritz and Leshem (2018) is also enabled by having players run a pre-defined Markov Chain together with signals indicated via collisions.
>
>   Thus, **all of the aforementioned approaches utilize collisions for implicitly signaling and communicating to some extent**. The major development in the recent works (Tibrewal et al., 2019; Boursier et al., 2020) is that such implicit communications are used for sharing arm statistics among players. However, this is again an algorithmic choice but not a violation of constraints. Moreover, there are a large (and growing) body of papers applying collision communications to the same decentralized MP-MAB problem. For example, many similar efforts have been made in this direction for the homogeneous setting (Boursier and Perchet, 2019; Wang et al., 2020; Shi et al., 2020). Even in the discussion of robustness (R1) or adversarial cases (Alatur et al., 2020; Shi and Shen, 2021), implicit communication is also widely adopted.
>
>   In summary, there do not exist two different settings as indicated by Reviewer 5qQj. The differences are purely from algorithmic choices. Furthermore, our problem formulation exactly follows previous works, and the adopted implicit communication is fundamentally possible because of the existence of collisions.
>
> - **Point 2 (value of the result)**: From results in distributed convex optimization, it is not surprising to have the regret close to the centralized setting.
>
>   **Ans**: We would like to (again) emphasize that in the field of heterogeneous MP-MAB, our result is the **first** to prove it is indeed possible to approach the centralized performance (although similar results might be standard in distributed convex optimization). Moreover, as shown in the paper, this is not a straightforward result to be obtained, and the ideas of incorporating CUCB and ADC are novel from our perspective.
>
>   Specifically, state-of-the-art results are listed in Table 1 in the paper. We can see that even using similar ideas of implicit communication and network-wise information processing (Boursier et al., 2020; Tibrewal et al., 2019), previous results are highly sub-optimal compared to BEACON, which is also verified in the experiments. Thus, we believe the current result and approach are valuable.
>
> - **Point 3 (contribution of the paper)**: The paper contribution is how to design a communication efficient scheme between the leader and each follower to achieve an "optimal" tradeoff between communication and system regret, instead of closing the "gap" between centralized and decentralized regrets.
>
>   **Ans**: Technically, this paper, especially the ADC protocol, is indeed about how to design a communication-efficient scheme and the communication-exploration-exploitation tradeoff, as indicated by the reviewer. However, such technical developments lead to the result of closing the gap between centralized and decentralized regrets, which is non-trivial from our perspective. We have consciously emphasized in lines 315--321, one of the future directions is how to remove the centralized reliance, but that does not devalue the current design and result.
>
> [R1] Boursier, E., & Perchet, V. (2020, July). Selfish robustness and equilibria in multi-player bandits. In Conference on Learning Theory (pp. 530-581). PMLR.

---

### Author Response · Authors · 2021-08-29
**Post rebuttal**

Dear reviewers:

Thank you for your insightful comments, and we hope that our responses have addressed your questions. If you have additional concerns or feedback after reading our responses, please do not hesitate to let us know, and we will be happy to answer them.

Thanks,

Authors of Paper 7433

---

### Decision · Program_Chairs · 2021-09-27

**Decision:**

Accept (Poster)

**Comment:**

This paper looks at the multi-player multi-arm bandit problem, with heterogenous rewards. This specific problem of reaching the centralized case with  decentralized protocols was still open, while it was settled with homogenous rewards.

This result requires combining additional, non trivial techniques from combinatorial bandits amongst others, hence hitting the acceptance bar.